# Neural Network Reparametrization for Accelerated Optimization in Molecular Simulations

**Nima Dehmamy**
IBM Research
Nima.Dehmamy@ibm.com

**Csaba Both**
Northeastern University
both.c@northeastern.edu

**Jeet Mohapatra**
MIT CSAIL
jeetmo@mit.edu

**Subhro Das**
IBM Research
subhro.das@ibm.com

**Tommi Jaakkola**
MIT CSAIL
tommi@csail.mit.edu

## Abstract

We propose a novel approach to molecular simulations using neural network reparametrization, which offers a flexible alternative to traditional coarse-graining methods. Unlike conventional techniques that strictly reduce degrees of freedom, the complexity of the system can be adjusted in our model, sometimes increasing it to simplify the optimization process. Our approach also maintains continuous access to fine-grained modes and eliminates the need for force-matching, enhancing both the efficiency and accuracy of energy minimization. Importantly, our framework allows for the use of potentially arbitrary neural networks (e.g., Graph Neural Networks (GNN)) to perform the reparametrization, incorporating CG modes as needed. In fact, our experiments using very weak molecular forces (Lennard-Jones potential) the GNN-based model is the sole model to find the correct configuration. Similarly, in protein-folding scenarios, our GNN-based CG method consistently outperforms traditional optimization methods. It not only recovers the target structures more accurately but also achieves faster convergence to the deepest energy states. This work demonstrates significant advancements in molecular simulations by optimizing energy minimization and convergence speeds, offering a new, efficient framework for simulating complex molecular systems. [1]

Scientific simulations, particularly in molecular dynamics (MD), face fundamental challenges in finding optimal configurations. The energy landscapes of these systems are characterized by numerous saddle points and local minima, making it difficult for traditional optimization methods to discover the most stable states. This complexity stems from the interplay between different scales of interactions, from strong covalent bonds to weak van der Waals forces, leading to slow convergence in gradient-based methods and often suboptimal results. For instance, in protein folding, the strong peptide bonds create steep energy barriers while weak hydrophobic interactions guide the overall folding process, creating a hierarchy of energy scales that is challenging to optimize simultaneously.

To address these challenges, coarse-graining (CG) methods have emerged as a popular approach, reducing computational complexity by decreasing the number of degrees of freedom (DOF) and clustering them into collective modes. While these methods have shown success (Pak & Voth, 2018; Hollingsworth & Dror, 2018), they face significant limitations. Traditional CG approaches require cumbersome procedures such as back-mapping (returning to the original DOF) and force-matching (finding the forces experienced by CG modes) (Jin et al., 2022), which can limit their efficiency and scalability. Moreover, the strict reduction of DOF in conventional CG can sometimes oversimplify the system, losing important fine-grained details necessary for accurate energy minimization.

---

[1] The code can be found at `https://github.com/nimadehmamy/coarse_graining_reparam`

38th Conference on Neural Information Processing Systems (NeurIPS 2024).

In this paper, we introduce an innovative alternative that overcomes these limitations through neural reparametrization. Instead of strictly reducing DOF as in conventional CG, our approach leverages an overparametrized neural ansatz to represent fine-grained (FG) modes as functions of CG modes. This reparametrization concept, similar to techniques such as Deep Image Priors (Ulyanov et al., 2018), enables the neural network to dynamically represent the FG system while maintaining continuous access to FG modes and eliminating the need for force-matching. The overparametrization provides additional flexibility in navigating the energy landscape - while the physical system has $n \times d$ degrees of freedom ($n$ particles in $d$ dimensions), our neural representation can use a higher-dimensional latent space to find paths around energy barriers that might be difficult to traverse in the original space.

A key innovation in our approach is the incorporation of Graph Neural Networks (GNN) with a structure informed by 'slow modes'–inherently stable collective modes identified through spectral analysis of the system's dynamics. We show that these modes arise naturally from the structure of physical Hessians, which are Laplacian matrices over particle indices for a broad class of potential energies. By focusing on these slow modes, which typically cause convergence bottlenecks in traditional optimization, we can significantly accelerate the learning process. The GNN architecture allows us to safely increase learning rates without stability issues, resulting in both faster dynamics progression and the discovery of lower energy states compared to direct optimization methods.

The effectiveness of our approach is demonstrated through experiments on both synthetic systems and real molecular structures. In particular, for protein folding with weak Lennard-Jones interactions, where traditional methods often struggle with the shallow energy landscape, our GNN-based model consistently finds deeper energy minima. This success can be attributed to two key factors: the ability of the overparametrized representation to explore the energy landscape more effectively, and the incorporation of physically meaningful slow modes into the neural architecture, which helps guide the optimization toward stable configurations.

The main contributions of this work are:

1. **CG via reparametrization:** A new paradigm that circumvents traditional challenges like force-matching and back-mapping.

2. **Robust slow modes:** Effective identification and utilization of stable modes across various systems.

3. **MD simulations:** Demonstrated improvements in efficiency and depth of energy exploration in protein dynamics.

4. **Overparametrization benefits:** Evidence that an overparametrized framework can outperform traditional DOF reduction in terms of convergence speed and energy minimization.

5. **Data-free optimization:** Our method modifies the optimization landscape without the need for training data, enhancing its applicability and efficiency.

# 1 Background

Traditional optimization in physics-based models, like (MD), faces unique challenges due to the shallow nature of these models, where physical DOF are the trainable weights. Additionally, the interactions occur at multiple scales, from strong covalent bonds to weak van der Waals forces, leading to slow convergence in gradient-based methods.

To address these challenges, conventional strategies include preconditioning with methods like adaptive gradient (Duchi et al., 2011; Kingma & Ba, 2014) or quasi-Newton (Fletcher, 2013), and CG, which simplifies the system by truncating DOF to focus on collective modes. However, both approaches have limitations: preconditioning methods struggle with cost and inefficacy due to non-diagonal Hessians in physics problems, and CG can be restrictive and require intensive back-mapping and force-matching steps (Jin et al., 2022).

In contrast, our approach utilizes neural network reparametrization to dynamically adjust system complexity, which may involve overparametrization. This method allows for flexible system representation, which can simplify the optimization process. It can help avoid local minima and accelerates convergence by exploring the configuration space more efficiently.

**Neural Reparametrization in Practice** Our neural reparametrization approach is not limited to reducing DOF but can also increase them when beneficial, offering an adaptive solution to the specific needs of a simulation. This flexibility is crucial for addressing the hierarchy of interactions in molecular systems, where different forces operate at vastly different scales.

**Implementation and Comparison to CG** While CG methods focus on predefined collective modes and often involve laborious optimization steps like force-matching and back-mapping, our neural reparametrization approach defines modes based on the spectrum of a canonical Hessian, directly incorporating these into the neural network's architecture. This not only bypasses the need for traditional CG steps but also enhances the adaptability and speed of the optimization process.

**Advantages Over Traditional Methods** Our method diverges from traditional data-driven ML approaches that require extensive datasets, which are often unavailable or costly to produce in molecular and material design. By not relying on training data, our approach provides a robust framework for tackling complex optimization problems, from molecular dynamics to protein folding, with improved efficiency and without the constraints of data availability.

## 1.1 Traditional Coarse-graining

Let $X \in \mathcal{X} \simeq \mathbb{R}^{n \times d}$ represent the degrees of freedom (DOF), such as particle positions or bond angles, and let $\mathscr{L} : \mathcal{X} \to \mathbb{R}$ denote the energy or potential function. The objective is to simulate the dynamics of the system or to find high-likelihood configurations $X^*$ that represent deep local minima of $\mathscr{L}$. Given that $n$ is typically large and $\mathscr{L}$ is a steep non-convex function, computations can be slow. Traditional coarse-graining (CG) maps $X$ to a reduced space of CG variables, $\mathcal{Z} \simeq \mathbb{R}^{k \times d}$, where $k \ll n$. Implementing dynamics using CG modes requires determining the inter-mode forces ("force-matching") and how to revert to $\mathcal{X}$ ("back-mapping").

**Force-matching.** The fine-grained (FG) energy function, $\mathscr{L}_{FG} : \mathcal{X} \to \mathbb{R}$, needs an approximate potential $\mathscr{L}_{CG} : \mathcal{Z} \to \mathbb{R}$ such that for $X \in \mathcal{X}$,

$$\text{CG:} \quad \phi : \mathcal{X} \to \mathcal{Z}, \quad \mathscr{L}_{CG}(\phi(X)) \approx \mathscr{L}_{FG}(X). \tag{1}$$

The process of finding $\mathscr{L}_{CG}$ is called force-matching, traditionally solved analytically but increasingly with machine learning for enhanced accuracy Jin et al. (2022); Majewski et al. (2023).

**Back-mapping.** The map $\mathcal{Z} \sim \mathbb{R}^{k \times d}$ is not unique, often resulting in multiple possible $X$ for a given $Z$. Back-mapping typically involves sampling or optimization to find physically plausible $X$ configurations, avoiding scenarios like overlapping atoms or high energies. This can be complex when many $X$ map to the same $Z$, with current methods ranging from geometric reconstruction Lombardi et al. (2016) to refinement with molecular dynamics Badaczewska-Dawid et al. (2020); Roel-Touris & Bonvin (2020) and data-driven approaches Yang & Gómez-Bombarelli (2023); Wang et al. (2022).

## 1.2 Neural Reparametrization as an Alternative to Coarse-graining

Instead of traditional CG, which reduces DOF through a mapping to a reduced space, our approach reparametrizes the DOF $X$ as a function of CG-like modes. This reparametrization, given by $X = \rho(Z)$, where $\rho : \mathcal{Z} \to \mathcal{X}$, offers a flexible, reversible mapping that inherently includes benefits such as direct access to fine-grained modes and elimination of force-matching and back-mapping needs:

$$\text{Reparametrization:} \quad X = \rho(Z), \quad \rho : \mathcal{Z} \to \mathcal{X} \tag{2}$$

1. **Flexible parametrization:** Leveraging neural overparametrization and architecture design.
2. **Direct access to fine-grained modes:** $X = \rho(Z)$ avoids the need for back-mapping.
3. **Simplified energy computation:** The energy for CG-like modes is $\mathscr{L}_{CG}(Z) = \mathscr{L}(\rho(Z))$.

While this method can be computationally intensive as $\mathscr{L}_{CG}(Z)$ is computed using $X$, the efficiency gains in optimization speed and depth of energy minimization can offset the costs.

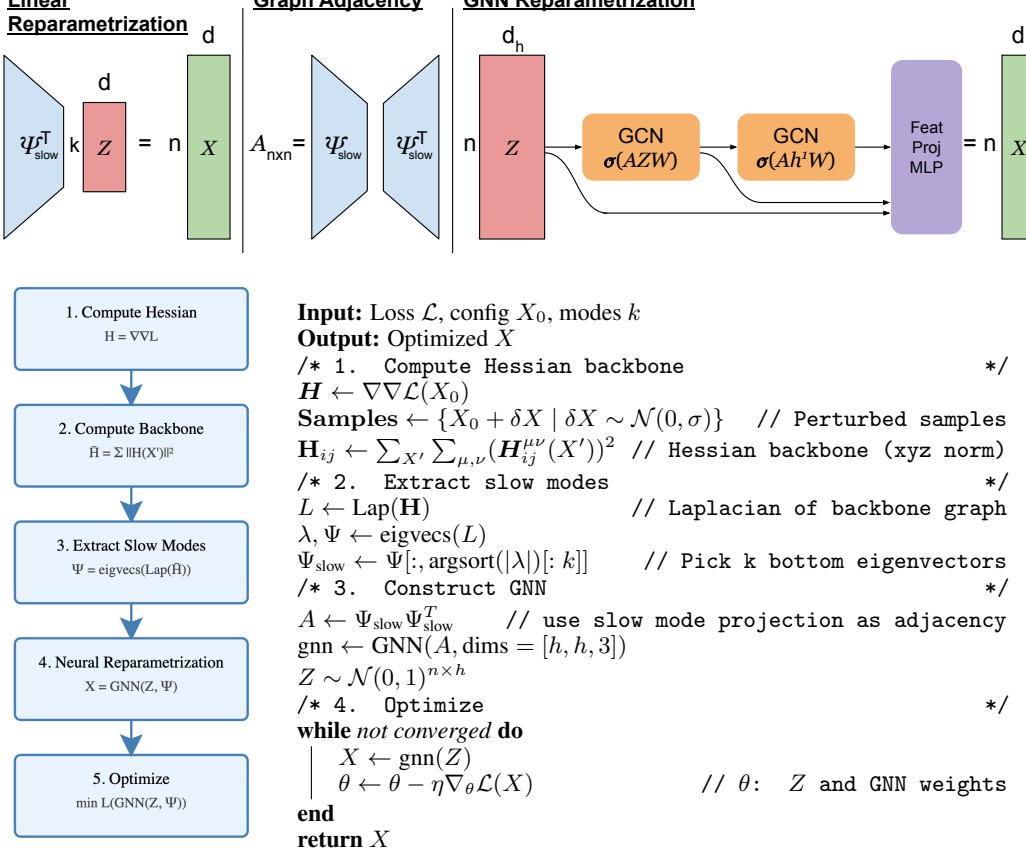

Figure 1: Overview of the neural reparametrization method. **Top:** Architectures used for reparametrization. In linear reparametrization, $X = Z^T \Psi_{\text{slow}}$. In the GNN case, we use the slow modes to construct a graph with adjacency $A = \Psi_{\text{slow}} \Psi_{\text{slow}}^T$ and use it in GCN layers to obtain $X = \text{gnn}(Z)$. **Left:** Flowchart showing the key steps of the method. **Right:** Detailed algorithm for implementation.

**Neural Architectures for Reparametrization** The reparametrization function $\rho$ can range from simple linear projections to complex neural networks. Initially, we employ a linear projection onto identified slow modes:

$$X = \rho(Z) = Z^T \Psi_{\text{Slow}} \equiv \sum_{i \in \text{Slow}} Z_i^T \psi_i \tag{3}$$

More generally, $\rho$ may be a deep neural network (DNN), similar to the approach taken in prior work suggesting neural priors (e.g. Deep Image Priors Ulyanov et al. (2018)).

**Graph Neural Networks (GNN) for Dynamic Reparametrization:** Extending beyond linear models, we explore the use of GNNs, inspired by recent advancements in graph-based optimizations Both et al. (2023). Here, the GNN reparametrizes node states and was shown to find both lower energy states and exhibit faster convergence. Our idea is to use a "Hessian backbone" as a graph, which acts as a weighted graph adjacency matrix for a GNN. In our experiments, we observe this GNN to have significant advantages over the direct as well as linear reparametrization equation 3. The details of our GNN architecture are discussed in Section 3. Next, we derive the properties of the slow modes for a large class of energy functions important in molecular systems.

### 1.3 The role of the Hessian

The success of optimization in molecular systems is fundamentally limited by the disparity in evolution rates along different modes of the system. Near any configuration $X$, the dynamics of

gradient-based optimization can be understood through the eigendecomposition of the Hessian $\boldsymbol{H} = \nabla\nabla\mathcal{L}$. The eigenvectors of $\boldsymbol{H}$ define the natural modes of the system, with their eigenvalues determining how quickly these modes evolve under gradient descent. Modes with large eigenvalues (fast modes) evolve rapidly but constrain the learning rate to ensure stability, while modes with eigenvalues close to zero (slow modes) evolve orders of magnitude more slowly, leading to extremely slow convergence, particularly near saddle points.

This disparity presents a fundamental challenge: To maintain numerical stability, the learning rate must be small enough to handle the fastest modes, but this makes the slow modes evolve at a glacial pace. Traditional approaches like adaptive gradient methods attempt to address this by approximating a diagonal preconditioner, but they struggle with the strongly coupled nature of physical systems where the Hessian is far from diagonal. While conventional coarse-graining partially addresses this by eliminating fast modes, it introduces other challenges such as force-matching and back-mapping.

Our approach takes a different perspective: instead of eliminating modes, we seek to identify and directly incorporate slow modes into our optimization process. However, this raises two key challenges. First, as the system evolves, the Hessian changes, potentially altering which modes are slow. Second, even if we can identify slow modes, we need a way to modify the optimization to preferentially explore these directions. The next section addresses the first challenge by proving that slow modes of physical Hessians are remarkably robust, arising from fundamental symmetries of the underlying interactions. We then show how these robust slow modes can be effectively utilized through neural reparametrization.

## 2 Properties of Physical Hessians

We will now show that the Hessian of potential energies important in physics and molecular systems enjoy certain properties that lead to the robustness of slow modes. In short, if we find a stable backbone for Hessians of different configurations $X$, then the slow modes of the Hessian at $X$ are close to the slow modes derived from the backbone.

**Invariant potentials.** In systems of interacting particles in physics, leading interactions are often pairwise and involve relative features, $\boldsymbol{r}_{ij} \equiv X_i - X_j$ (distance vector, relative angle, etc). These interactions are invariant under global symmetries, such as Euclidean symmetries (translation and rotation) or Lorentz symmetry (relativistic particles). These symmetries maintain the invariance of certain norms, $v^2 = \|\boldsymbol{v}\|_\eta \equiv \boldsymbol{v}^T \eta \boldsymbol{v}$, where $\eta$ may be the Euclidean metric $\eta = \text{diag}(1,1,1)$ or the Minkowski metric $\eta = \text{diag}(-1,1,1,1)$. For example, the Euclidean norm $\boldsymbol{v}^T \boldsymbol{v}$ in $d$ dimensions is invariant under rotations $\boldsymbol{v} \to g\boldsymbol{v}$, where $g \in SO(d)$.

**Energy function structure.** Let $r$ denote the matrix of distances with $r_{ij} = \|\boldsymbol{r}_{ij}\|_\eta$. Any function of $r_{ij}$ is invariant under symmetries that keep $\|\cdot\|_\eta$ invariant. Assuming additivity, the energy function can be written as:

$$\mathcal{L}(X) = \sum_{ij} f_{ij}(r_{ij}) \tag{4}$$

where $f_{ij}(z) = f_{ji}(z)$. For example, the Coulomb potential between particles $i$ and $j$ with charges $q_i$ and $q_j$ respectively, is given by $f_{ij}(z) = kq_i q_j/z$. The Lennard-Jones potential $f_{ij}(z) = A_{ij}/z^{1}2 - B_{ij}/z^6$ in molecular systems is also of this form.

### 2.1 Hessian of invariant potentials

The Hessian of potentials of the form equation 4 has the special property that it is the graph Laplacian of a weighted graph which depends on $X$, as we show now (see appendix E for details). This will play a crucial role in our argument about the robustness of the slow modes.

**Hessian as a graph Laplacian.** Recall the Laplacian of an undirected graph with adjacency matrix $A$ is defined as $L = \text{Lap}(A) = D - A$, where $D$ is the degree matrix with elements $D_{ij} = \delta_{ij}\sum_k A_{ik}$. The components of Laplacian can also be written as $L_{ij} = \sum_k A_{ik}(\delta_{ij} - \delta_{jk})$. We show that the Hessian of $\mathcal{L}$ in equation 4 is a Laplacian. Let $\partial_i \equiv \partial/\partial X_i$ and let $\hat{r} = \eta\boldsymbol{r}/r$ be the dual unit vector of $\boldsymbol{r}$. First, observe that $\partial_i r_{jk} = \hat{r}_{jk}(\delta_{ij} - \delta_{ik})$ where $\hat{r}_{jk}$ is the unit vector

of $\boldsymbol{r}_{jk}$ and $\delta_{ij}$ is the Kronecker delta (1 if $i = j$, 0 otherwise). Let $\mathrm{Hes}[g]$ denote the Hessian of a function $g$. We find that (app. E)

$$\mathrm{Hes}[\mathscr{L}](X)_{ij} = \partial_i \partial_j \mathscr{L}(X) = \sum_k (\delta_{ij} - \delta_{jk}) \boldsymbol{H}_{ik}(X) = \mathrm{Lap}(\boldsymbol{H})_{ij} \tag{5}$$

where $\boldsymbol{H}_{ik}(X) = \mathrm{Hes}[f_{ik}](r_{ik})$. Note that $\boldsymbol{H}$ has four indices, with components $\boldsymbol{H}_{ij}^{\mu\nu}$, having two particle indices $i, j$ and two spatial indices $\mu, \nu$. Thus, for every pair of spatial indices $\mu, \nu$, the Hessian $\boldsymbol{H}^{\mu\nu}$ is a Laplacian over particle indices. The Hessian being Laplacian has an important effect on its null eigenvectors. To show this we make use of the incidence matrix.

We are interested in the eigenvalues and eigenvectors of $\mathbf{H}$, as these characterize the slow and fast modes of the system. First, given a weighted adjacency matrix $A$ of a graph, let $\hat{A}$ and $\hat{L}$ be the "unweighted" adjacency and Laplacian matrices, where $\hat{A}_{ij} = 1$ if $A_{ij} \neq 0$ and zero otherwise. It follows that the null spaces of $L$ and $\hat{L}$ are shared:

**Theorem 2.1** (Null Space of the Laplacian). *Let $\mathbf{Null}[M]$ denote the null space of a symmetric real matrix $M$. The null space of the unweighted Laplacian $\hat{L}$ is contained within the null space of the weighted Laplacian $L$, i.e., $\mathbf{Null}[\hat{L}] \subseteq \mathbf{Null}[L]$.*

*Sketch of proof.* For any vector, $\boldsymbol{v} \in \mathbb{R}^n$, $\boldsymbol{v}^T \mathrm{Lap}(A) \boldsymbol{v} = \sum_{ij} A_{ij}(v_i - v_j)^2$. Since $\hat{A}_{ij} = 0$ yields $A_{ij} = 0$, but not necessarily vice versa, null vectors of $\mathbf{Null}[\hat{L}] \subseteq \mathbf{Null}[L]$. See appendix for full proof. $\qquad\square$

**Definition 2.1** (Slow manifold). *Let $L$ be a graph Laplacian (undirected, weighted or unweighted), with spectral expansion $L = \sum_{i=1}^n \lambda_i \psi_i \psi_i^T$. Let $\varepsilon \ll 1$ and $\lambda_{\max} = \max\{\lambda_i\}$ be the largest eigenvalue of $L$. We define the slow manifold as*

$$\mathbf{Slow}_\varepsilon[L] = \mathrm{Span}\{\psi_i \,||\lambda_i| < \varepsilon^2 \lambda_{\max}\} \tag{6}$$

**Theorem 2.2** (Slow modes of weighted Laplacians). *Let $A$ be the adjacency matrix of a weighted graph and $\hat{A}$ be its unweighted counterpart. Let $L = \mathrm{Lap}(A)$ and $\hat{L} = \mathrm{Lap}(\hat{A})$. Then $\mathbf{Slow}_\varepsilon[L]$ overlaps with $\mathbf{Slow}_\varepsilon[\hat{L}]$ up to $O(\varepsilon^2)$ corrections from the rest of the modes.*

The sketch of the proof relies on relating the spectra of the weighted and unweighted Laplacians using the incidence matrix $C$, as $L = \frac{1}{2} C W C^T$ and $\hat{L} = \frac{1}{2} C C^T$. For a random configuration $X$ the edge weights $W$ will be random, as they arising from derivatives of $f_{ij}(r_{ij})$ in equation 20 (unless $f_{ij}$ is quadratic which makes $W$ constant). Then, using the assumption of randomness on the weights $W$, we can show the slow modes of $L$ are perturbations of order $\varepsilon^2$ on slow modes of $\hat{\mathscr{L}}$. See Appendix D for proof.

**Implications for Coarse-Graining.** The identification of slow modes in the Hessian is crucial for coarse-graining, as these modes capture the essential dynamics of the system at larger scales. By focusing on these slow modes, we can develop reduced models that retain the key physical properties while being computationally more efficient.

**Coarse-Graining via Slow Modes.** The identification of slow modes in the Hessian enables an effective coarse-graining approach, where fast dynamics are averaged out, retaining only the slow, relevant dynamics. This method is particularly advantageous in reducing computational complexity while preserving critical structural information.

## 2.2 Hessian Backbone and Robust Slow Modes

The slow modes of the Hessian $\mathrm{Hes}[\mathscr{L}](X) = \mathrm{Lap}(\boldsymbol{H}(X))$ can dynamically change during optimization. To ensure the robustness of these modes, we need a proxy for the unweighted adjacency matrix $\hat{A} \equiv \mathbf{H}$. To this end, we aggregate Hessians from perturbed configurations $\mathbf{Samples}(X) = \{X' = X + \delta X\}$:

$$\textbf{Backbone:} \quad \mathbf{H}_{ij} = \sum_{X' \in \mathbf{Sample}(X)} \|H_{ij}(X')\|^2 \tag{7}$$

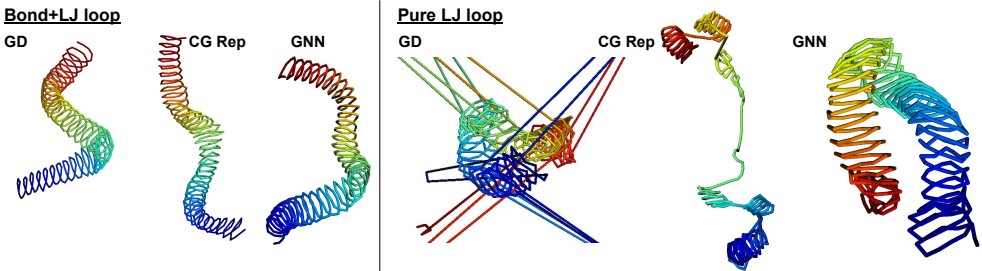

Figure 2: **Synthetic loop experiments.** Example runs of the synthetic loop experiments with $n = 400$ nodes. On the left (Bond+LJ), the potential is the sum of a quadratic bond potential $E_{bond}$ and a weak LJ (12,6) $E_{LJ}$. The bonds form a line graph $A_{bond}$ connecting node $i$ to $i + 1$, and a 10 weaker $A_{loop}$ connecting node $i$ to $i + 10$ via the LJ potential. To the right (Pure LJ) where the interactions are all LJ, but with a coupling matrix $A = A_{bond} + 0.1A_{loop}$. In Bond+LJ, GD already finds good energies and the configuration is reasonably close to a loop, though flattened. Both linear CG reparametrization (CG Rep) and GNN also find a good layout. The pure LJ case is much more tricky. But in most runs, GD almost gets the layout, but some nodes remain far away. The CG Rep fails to bring all the pieces together. Only GNN succeeds in finding the correct layout.

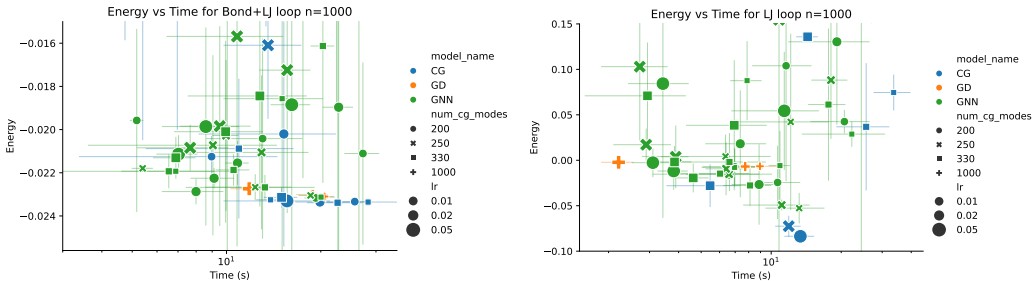

Figure 3: **Synthetic loop folding** ($n = 1000$). Lower means better for both energy and time. In Bond+LJ (left), a quadratic potential $\sum_i (r_{ii+1} - 1)^2$ attracts nodes $i$ and $i + 1$. A weak LJ potential attracts nodes $i$ and $i + 10$ to form loops. In LJ loop (right) both the backbone $i, i + 1$ and the 10x weaker loop are LJ. Orange crosses denote the baseline GD, green is GNN and blue is CG. The dots are different hyperparameter settings (LR, Nr. CG modes, stopping criteria, etc.) with error bars over 5 runs. In Bond+LJ, CG yields slightly better energies but takes longer, while GNN can converge faster to GD energies. In pure LJ, using CG and GNN can yield significantly better energies.

This aggregation helps identify consistently significant components across configurations, aiding in the extraction of reliable slow modes that remain effective over extended periods of optimization. In equation 7, $i, j \in \mathbb{Z}_n$ are the particle indices and the Frobenius norm $\|H_{ij}\|^2 = \sum_{\mu,\nu}(H_{ij}^{\mu\nu})^2$ sums over the feature indices (note that $X_i^{\mu}$ has a particle index $i$ and a feature index $\mu \in \{1, \ldots d\}$). Then, we extract the slow modes of the backbone, by doing a spectral expansion $\mathbf{H} = \sum_i \lambda_i \psi_i \psi_i^T$ and picking $\psi_i$ with $|\lambda_i| < \varepsilon^2 \max_j[\lambda_j]$, for some small $\varepsilon < 1$. The intuition behind equation 7 is to identify the components in the sampled Hessians which have consistently high magnitudes. If we had taken a simple mean we could get very small values, because the components can fluctuate randomly. Also, if we had taken the variance instead of the norm, we would get zero for quadratic $\mathscr{L}$, where $H$ is constant and has no variance. As we discussed above, the slow modes of the backbone $\mathbf{H}$ approximate the slow modes of sampled $H(X')$ up to $O(\varepsilon^2)$ errors.

## 3   Experiments

We apply our method to protein folding using classical MD forces.
**Settings:** We use gradient descent to minimize $\mathscr{L}(X)$. All experiments (both CG and baseline) use the Adam optimizer with a learning rate $10^{-2}$ and early stopping with $|\delta\mathscr{L}| = 10^{-6}$ tolerance and 5 steps patience. We ran each experiment four times.
**Baseline:** we use gradient descent (GD) with Adam optimizer on the MD energy as baseline.

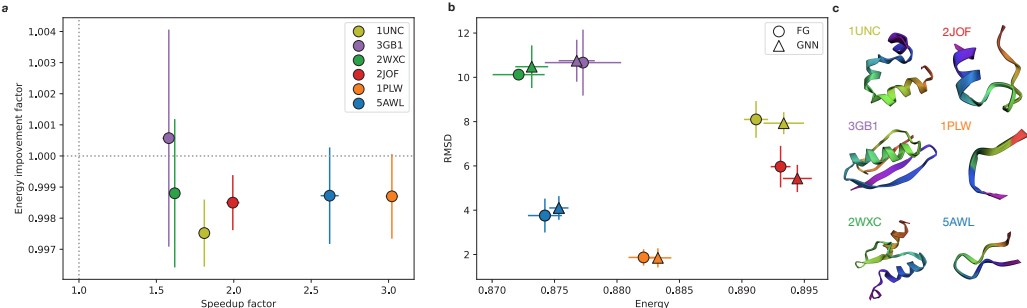

Figure 4: **Protein folding simulations** Figure (a) shows the energy improvement factor (FG energy / GNN energy) in the function of the speedup factor (FG time / GNN time) for the six selected proteins marked with different colors (c). In all cases, the GNN parameterization leads to speed improvement while it converges higher energy. (b) However, the higher energy in some cases, 2JOF and 1UNC proteins, results in a slightly lower RMSD value, which measures how close the final layout is to the PDB layout. The data points are averaged over ten simulations per protein.

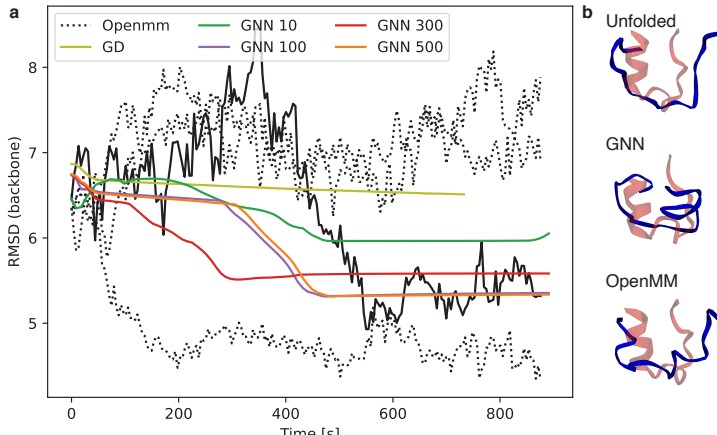

Figure 5: **2JOF (Trp-Cage) protein folding.** Figure (a) shows the RMSD value evolution of the 2JOF protein as it goes from an unfolded to a folded stage. At every step, we calculated the RMSD of the current layout compared to the PDB layout. We ran the OpenMM simulations at $298K$ temperature with 2fs timestep for 800000 steps, while the GNN and GD simulations were performed for 400000 steps with various hidden dimensions (10, 100, 300, 500). The black curves show the stochastic nature of protein folding using OpenMM. (b) The first figure shows the PDB (red) and unfolded (blue) layout; the second one is the GNN 500 final layout (blue), while the third is one of the OpenMM layouts, corresponding to the black curve.

**CG model:**    We use four different choices for the fraction of the eigenvectors to use in CG equation 3: $3 \times (\#\text{AminoAcids})$, 30%, 50%, and 70%. We use a two stage process. First, we use CG as in equation 3 $X = \rho(Z) = Z^T \Psi_{\textbf{Slow}}$ and minimize $\mathscr{L}_{CG}(Z) = \mathscr{L}(\rho(Z))$ over $Z$. After convergence to $X_0 = \rho(Z_0)$, we add $\delta X$ to $X_0$ and optimize the fine-grained $\delta X$, starting with $\delta X = 0$.

**GNN model:**    We use a GNN consisting of a graph convolution (GCN) layer with self-loops and one node-wise MLP layer, projecting the GNN output to 3D to get particle positions. The GCN takes $Z_{h_0} \in \mathbb{R}^{n \times h_0}$ as input, with $h_0 > 3$ and has weights $W_G \in \mathbb{R}^{h_0 \times h_1}$. Then, GCN output gets a Tanh activation and is passed to the MLP layer to yield $X$. The CG parameters in this case are $Z_h, W_G$ and the weights and biases of the MLP.

**Synthetic coil:**    We use quadratic and LJ potentials to make synthetic systems whose minimum energy state should be a coil (looping every 10 nodes), inspired by MD potentials. Figure 3 shows many experiments using GD, CG, and GNN. In the quadratic Bond+LJ case, GNN yields a good

**Robustness of GNN results**

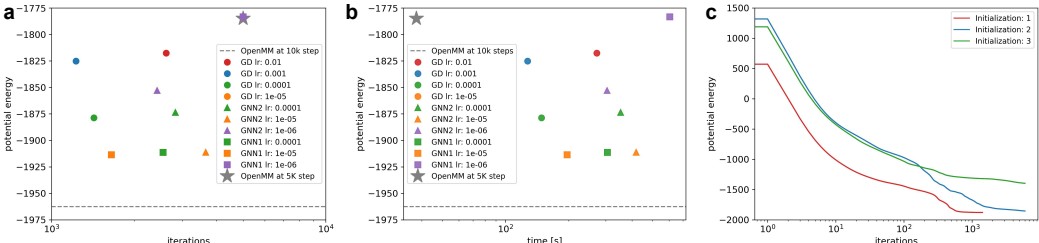

Figure 6: **Learning rate and initialization in protein folding for pdb 2JOF:** We conducted a sweep of the learning rate to see how robust the advantage of GNN over direct GD is. In **a** and **b** we show the energy achieved by GD and GNN vs the number of iterations and wallclock time. GNN1 and GNN2 use one and two GCN layers, respectively. We used early stopping which generally stopped the runs after 3-5k steps. The grey star shows the OpenMM results after 5k steps, which has a worse (higher) energy than our GD and GNN runs, but it takes a fraction of the time (it has many efficiency tricks that our code doesn't have). The dashed line shows the energy achieved by OpenMM after 10k steps. As we see, some of our GNN models reach energies close to the 10k steps of openMM in a fraction of the steps. All experiments show the best energy among three runs. **c** shows the effect of initialization on the GD runs. We do find the protein converges to significantly different conformations based on the init.

speedup, while CG yields better energies. The benefit of CG and GNN become more apparent in the harder pure LJ problem, where GD fails to find good energies, while CG finds much deeper energies, followed by GNN (Fig. 2).

**Protein folding with classical MD:** We implement a simplified force-field with implicit solvent (i.e. water molecules are not modeled and appear as hydrogen-bonding and hydrophobicity terms; app. A). In protein folding our energy function consists of five potential energies: bond length $E_{bond}$, bond angles $E_{angle}$, van der Waals $E_{vdW}$, hydrophobic $E_{hp}$ and hydrogen bonding $E_H$ Ceci et al. (2007). Figure 7 shows an example of these coupling matrices for the Enkephalin (1PLW) protein. To evaluate the effect of our CG model, we run experiments on four small proteins: Chignolin (5AWL), Trp-Cage (2JOF), Cyclotide (2MGO) and Enkephalin (1PLW).

**Protein Folding with Classical MD Using AMBER Force Field** In the updated simulation approach, we incorporate the AMBER force field, known for its accurate representation of molecular interactions, particularly in proteins. This force field is implemented using the parameters from OpenMM Eastman et al. (2017), and it comprehensively models the following interactions:

- Bond lengths $E_{bond}$ and bond angles $E_{angle}$
- Torsional angles $E_{torsion}$
- Non-bonded interactions including van der Waals $E_{vdW}$ and electrostatic $E_{elec}$ forces

We utilize the functional forms and parameters specified in the AMBER force field:

$$E_{bond} = \sum_{bonds} k_{bond}(r - r_0)^2 \qquad E_{angle} = \sum_{angles} k_{angle}(\theta - \theta_0)^2 \qquad (8)$$

$$E_{torsion} = \sum_{torsions} V_n \left[1 + \cos(n\omega - \gamma)\right] \qquad E_{vdW} = \sum_{i<j} \frac{A_{ij}}{r_{ij}^{12}} - \frac{B_{ij}}{r_{ij}^6} \qquad (9)$$

$$E_{elec} = \sum_{i<j} \frac{q_i q_j}{4\pi\epsilon_0\epsilon_r r_{ij}} \qquad (10)$$

Here, $r$ and $\theta$ represent the bond lengths and angles, respectively, with $r_0$ and $\theta_0$ as their equilibrium values. The torsional term $E_{torsion}$ includes a sum over all torsion angles $\omega$, with periodicity $n$,

amplitude $V_n$, and phase $\gamma$. The Lennard-Jones potential in $E_{vdW}$ is characterized by parameters $A_{ij}$ and $B_{ij}$, and $E_{elec}$ is calculated using the Coulombic potential with partial charges $q_i$, $q_j$ and the relative permittivity $\epsilon_r$.

In this simulation, we exclude the modeling of solvent effects entirely, focusing solely on the protein in vacuum. This approach simplifies the computational model while emphasizing the direct interactions within the protein.

The overall energy of the system is then given by:

$$\mathcal{L}(X) = E_{bond} + E_{angle} + E_{torsion} + E_{vdW} + E_{elec} \tag{11}$$

Figure 7 shows the interaction matrices for the Enkephalin (1PLW) protein. Our framework has been extended to efficiently compute these energies and gradients, facilitating the simulation of protein folding dynamics in our coarse-grained model. We test our model on several small proteins including Chignolin (5AWL), Trp-Cage (2JOF), Cyclotide (2MGO), and Enkephalin (1PLW) to evaluate the effectiveness of our approach.

**Protein results:** Denoting the final energy and run time of the GNN model by $E_{GNN}$ and $t_{GNN}$, and baseline by $E_0$ and $t_0$, we compute the energy improvement factor $\delta\hat{E} = E_0/E_{GNN}$ and speedup factor $\delta\hat{t} = t_0/t_{GNN}$, to plot different proteins together. Figure 4a shows the mean of $\delta\hat{E}$ vs $\delta\hat{t}$ over the 10 runs for GNN the model with hidden dimensions 300 (error bars are 1 STD). Overall, we find that all GNN models outperform the baseline in terms of run time and, eventually, also with energy improvement. To measure the folding quality, we use RMSD, comparing the final layouts to the PDB structure.

Figure 5 shows the RMSD value evolution using different methods. While, in most cases, OpenMM reaches a deeper RMSD value, our models could serve as a good initializer for accelerating molecular dynamics. To evaluate the robustness of these results, we ran sweeps over the learning rate, varied the number of GNN layers (one or two layers), and varied the initialization.

Figure 6 shows the results of these tests for the protein 2JOF. We used early stopping for switching from CG to FG in our GNN models and GD, resulting in 3-5k iteration steps. Compared with 5k steps of OpenMM simulations, both our GNN models and GD with Adam reach significantly deeper energies with fewer steps (a), with the lowest energies being all GNN. However, OpenMM takes less wall-clock time (b). Nevertheless, the depth of the energies achieved by GNN at 3-5k steps is close to 10k steps with OpenMM. More efficient implementations of our GNN may further improve these results.

## 4   Discussion

We showed preliminary evidence that CG through reparametrization can yield some improvements over non-CG baseline in protein folding, both in terms of run time as well as energy. This method has the advantage that it does not require force-matching or back-mapping. However, more experiments are needed to compare it against traditional CG methods. In fact, using ML to learn force-matching might provide further advantage by removing the need to evaluate $\mathscr{L}_{CG}(Z) = \mathscr{L}(X)$ via the fine-grained modes $X$. Also, while our canonical slow modes are derived for physical Hessians, the reparametrization approach to CG is general and could be applied to other ML problems.

## Acknowledgment

JM was partly supported by a grant from the MIT-IBM Watson AI Lab. CB's work was done partly during his internship at the MIT-IBM Watson AI Lab.

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

## A    Protein folding with classical MD

In protein folding our energy function consists of five potential energies for: bond length $E_{bond}$, bond angles $E_{angle}$, Van der Waals $E_{vdW}$, hydrophobic $E_{hp}$ and hydrogen bonding $E_H$ Ceci et al. (2007). Note that we are ignoring the solvent (e.g. water) and writing using potentials, or force fields. To calculate the force field, we use distance, $r$, and angle-based, $\Theta$, potentials. For each amino acid, we use the rdkit Landrum et al. (2020) package to acquire bond length, $r_0$, and bond angle, $\theta_0$ (every triplet of atoms defining the bond), information that we use to define quadratic energies $E_{bond}$ and $E_{angle}$. We use Lennard-Jones (LJ) potentials, $V_{p,q}(r) = r^{-p} - r^{-q}$, to approximate $E_{vdW}$ between all pairs of atoms, $E_H$ between atoms prone to form a hydrogen bond (certain $H$ and $O$, in our case), $E_{hp}$ between atoms in hydrophobic residues, yielding

$$
\begin{aligned}
\mathscr{L}(X) =& E_{bond} + E_{angle} + E_{vdW} + E_H + E_{hp} \\
=& k_{bond}(r - r_0)^2 + k_{angle}(\theta - \theta_0)^2 \\
& + \epsilon_{vdW} V_{12,6}\left(\frac{r}{\sigma_{vdW}}\right) + \epsilon_H V_{6,4}\left(\frac{r}{\sigma_H}\right) + \epsilon_{hp} V_{6,4}\left(\frac{r}{\sigma_{hp}}\right)
\end{aligned}
\tag{12}
$$

Here the coupling matrix $[\sigma_{vdW}]_{ij} = a_i + a_j$ where $a_i$ is the vdW radius of atom $i$. For atoms which form H-bonds, $[\sigma_H]_{ij} = (b_i \cdot b_j)1.5\text{Å}$ (hydrogen bonding radius) with $b_i = 1$ if $i$ forms an H-bond, and $b_i = 0$ otherwise. $[\sigma_{hp}]_{ij} = c_i + c_j$ where $c_i = 2\text{Å}$ if atom $i$ is in a hydrophobic residue and $c_i = 0$ otherwise.

We note that our choices for $\epsilon_H, \epsilon_{vdW}, \epsilon_{hp}$ and $k_{bond}, k_{angle}$, can be a source of error. Additionally, we "softened" the LJ potential to $V_{p,q} = 1/(r^p + \zeta) - 1/(r^q + \zeta)$ with $\zeta = 0.65$, which is large and significantly reduces the penalty for overlapping atoms and may reduce accuracy.

## B    Additional Figures

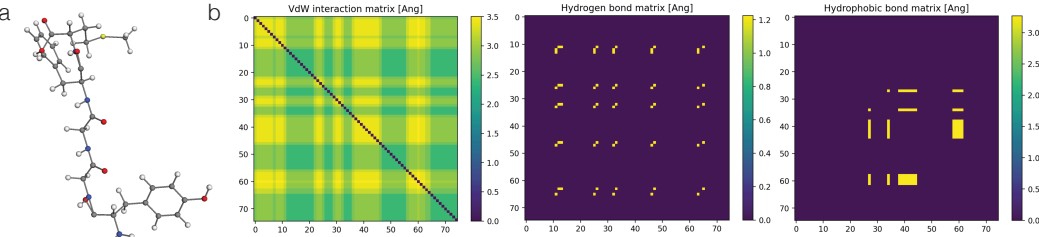

Figure 7: Enkephalin (1PLW). a) The peptide chain is built by stacking amino acids on each other using the peptide bond length from the literature, $1.32$ Å. b) Van der Waals, hydrogen bond, and hydrophobic interaction matrix, that we use in the energy optimization.

## C    Energy minimization

Let $X \in \mathcal{X} \simeq \mathbb{R}^{n \times d}$ be a set of degrees of freedom (e.g. particle positions, bond angles, etc.) and let $\mathscr{L} : \mathcal{X} \to \mathbb{R}$ be the energy (loss) function. We are interested in finding configurations $X^*$ which are local minima of $\mathscr{L}$. We can find such $X^*$ using a gradient descent (GD), or its continuous variant, gradient flow (GF)

$$
\frac{dX}{dt} = -\varepsilon \boldsymbol{\nabla} \mathscr{L}(X)
\tag{13}
$$

where $\varepsilon$ is the matrix of learning rates (LR). In simple GD where $\varepsilon = cI$ is a single constant times identity, GD evolves at different rates in different directions, with some being much slower than others. At a given $X$, these "slow modes" are the eigenvectors of the Hessian $H(X) = \boldsymbol{\nabla}\boldsymbol{\nabla}\mathscr{L}(X)$ with eigenvalues closest to zero, as we review below. We will first define fast and slow modes in the simple quadratic case and then generalize them to non-convex cases in the next section.

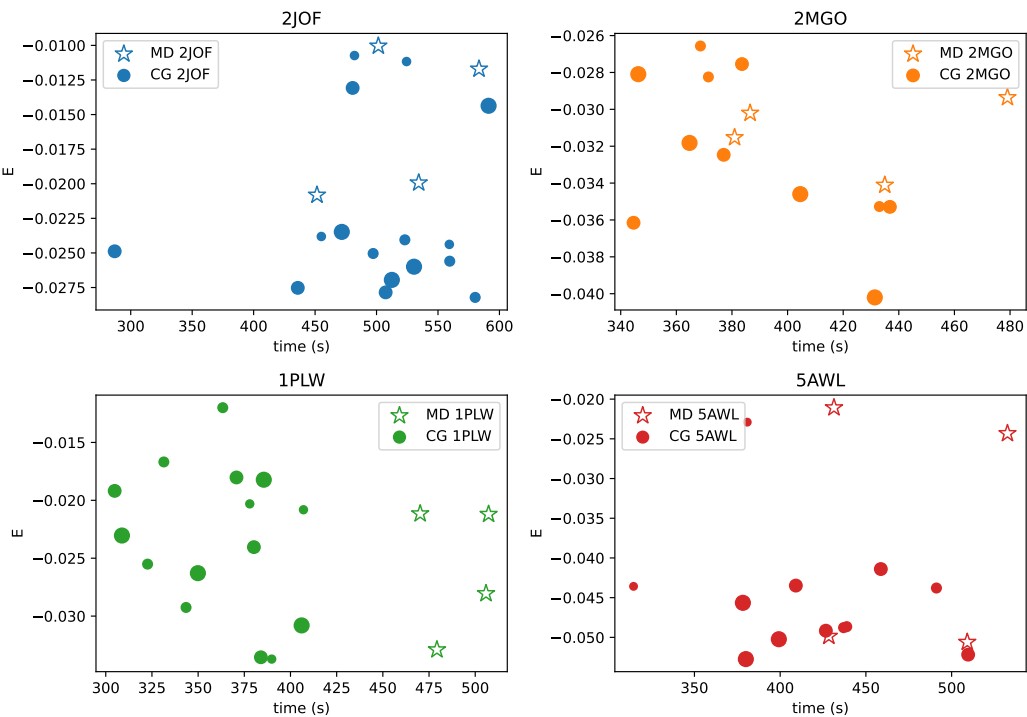

Figure 8: Comparison of performance of CG Hessian versus baseline MD. Point sizes correspond to the number of CG modes used.

**Fast and slow modes for quadratic Loss.** Consider the case where $\mathscr{L}(X) = \frac{1}{2}\operatorname{Tr}\{X^T \boldsymbol{H} X\}$. Here $\boldsymbol{H}$ is a Hermitian matrix and the Hessian of $\mathscr{L}$, with a spectral expansion given by $\boldsymbol{H} = \sum_i \lambda_i \psi_i \psi_i^T$, $\lambda_i \in \mathbb{R}$ and $\psi_i \in \mathbb{R}^n$. In this basis we have $X(t) = \sum_i c_i(t)\psi_i$ with $c_i : \mathbb{R} \to \mathbb{R}^d$. Projecting equation 13 onto one of the eigenmodes we get

$$\frac{dc_i}{dt} = \psi^T \frac{dX}{dt} = -\varepsilon\lambda_i \psi^T X = -\varepsilon\lambda_i c_i \tag{14}$$

where we assumed $d\psi_i/dt = 0$. From equation 14 we see that the decay/growth rate along mode $\psi_i$ is $|\varepsilon\lambda_i|$. Hence, modes with $\lambda_i$ close to zero are the "slow modes", evolving very slowly, and large $|\lambda_i|$ defines the "fast modes". Since $c_i(t) = c_i(0)\exp[-t/\tau_i]$ with time scale $\tau_i = 1/(\varepsilon\lambda_i)$, the fast modes evolve exponentially faster than slow modes. This disparity in the rates results in slow convergence, because the fast modes force us to choose smaller $\varepsilon$ to avoid numerical instabilities. Two potential ways to fix the issue with disparity in time scales are: 1) make rates isotropic (second-order methods and adaptive gradients); 2) mode truncation or compression (CG). We will briefly review the former here.

**Adaptive gradient and second-order methods.** Newton's method uses $\varepsilon = \eta H(X)^{-1}$ which makes GD isotropic along all modes, but it is expensive ($O((3n)^3)$ in our case). Quasi-Newton methods, e.g. BFGS Fletcher (2013), approximate $H^{-1}$ iteratively, but are generally also slow. Another, more efficient approach is adaptive gradient methods, such as AdaGrad Duchi et al. (2011) and Adam Kingma & Ba (2014) which approximate $H$ by $\sqrt{g_t g_t^T + \eta}$ where $g_t = \sum_{i=1}^k \gamma^i \boldsymbol{\nabla}\mathscr{L}(X(t-i))$ is some discounted average over past gradients and $\eta$ a small constant. For efficiency, in practice we only use the diagonal part of this matrix to approximate $H^{-1}$. As we will see in experiments, this approximation, while being far superior to GD with constant LR, is still very slow for MD tasks.

Most second-order methods are designed to work for generic problem and don't make strong assumptions about the spectrum of the Hessian. Recent second-order methods such as K-FAC Martens & Grosse (2015) and Shampoo Gupta et al. (2018) work with block diagonal approximations of the Hessian (or the Fisher information matrix), which usually emerges in deep learning models

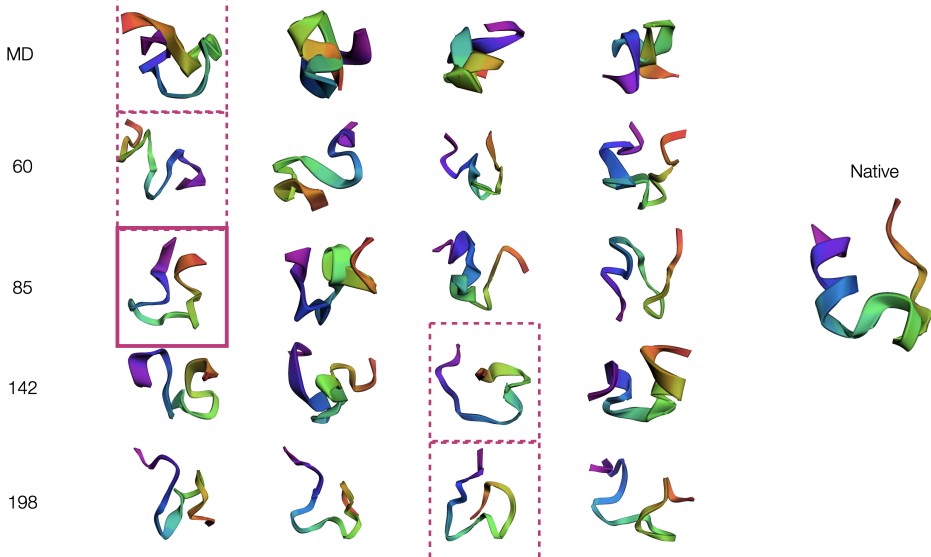

Figure 9: The folded structures of the 2JOF protein by using the CG and baseline method. The numbers in front of the rows are the numbers of eigenvectors used in the CG reparametrization. Dashed frames show the minimum energy embedding in each case, while the thick line frame highlights the absolute minimum layout.

due to model architecture. Instead, we will exploit the spectral properties of the Hessian in physics problems. Fast and slow modes generally arise in physics due to vastly different strengths in forces (e.g. weak van der Waals vs strong chemical bonds).

### C.1 Generalized fast and slow modes

The notion of fast and slow modes is helpful for the analysis of any time slice of the dynamics during which the Hessian is not changing dramatically. Consider a configuration $X(t)$ and let $\delta t$ be a small time interval. We are looking for modes which are almost stationary over $\delta t$. To identify these modes, we can for instance find perturbations $\delta X$ which would have almost zero dynamics. concretely we find the dynamics of $X + \delta X$ as

$$\frac{d}{dt}(X + \delta X) = -\varepsilon \boldsymbol{\nabla}\mathscr{L}(X + \delta X) \approx -\varepsilon \boldsymbol{\nabla}\mathscr{L}(X) - \varepsilon H \delta X + O(\delta X)^2 \tag{15}$$

meaning, a small $\delta X$ adds $\varepsilon H \delta X$ to the dynamics.

Thus if $\delta X$ is a zero mode of the Hessian, $H\delta X = 0$, it won't change the dynamics of $X$. To define slow modes, we can slightly relax this and look for normalized modes $\psi = \delta X / \|\delta X\|$ whose associated time scale is much longer than a desired time scale $\delta t$

$$\tau = |\varepsilon \psi^T H \psi| = |\varepsilon \lambda| \gg \delta t \tag{16}$$

which just means that we need to find the approximate zero modes of the Hessian $H(X)$.

**CG by projecting to the slow manifold.** Because the dynamics of the modes above is very slow over $\delta t$, we can safely increase the time scale and run their dynamics over much longer periods $\Delta t \gg \delta t$. The essence of our algorithm is to ignore fast modes and project and evolve the system on the "slow manifold" spanned by the slow modes of the Hessian. However, the main challenge is how to deal with the fact that the Hessian is not constant and depends on the configuration $X$. We address this point next. We show that for a large class of physical potentials one can find a reliable set of approximate slow modes.

# D   Properties of Physical Hessians

**Invariant potentials.**   In systems of interacting particles in physics, most of the leading interactions are pairwise and involve relative features, $\boldsymbol{r}_{ij} \equiv X_i - X_j$ (distance vector, relative angle, etc). Moreover, they are often invariant under certain global symmetries, such as Euclidean symmetries (translation and rotation) or Lorentz symmetry (relativistic particles). These symmetries keep some 2-norm of vectors, $v^2 = \|\boldsymbol{v}\|_\eta \equiv \boldsymbol{v}^T \eta \boldsymbol{v}$ invariant. Here $\eta$ may be the Euclidean metric $\eta = \mathrm{diag}(1,1,1)$ or the Minkowski metric $\eta = \mathrm{diag}(-1,1,1,1)$ for relativistic problems, etc. For example, the Euclidean norm $\boldsymbol{v}^T \boldsymbol{v}$ in $d$ dimensions is invariant under rotations $\boldsymbol{v} \to g\boldsymbol{v}$, where $g \in SO(d)$, and the Minkowski norm is invariant under the Lorentz group $SO(1, d-1)$.

Let $r$ denote the matrix of distances with $r_{ij} = \|\boldsymbol{r}_{ij}\|_\eta$. Any function of $r_{ij}$ is invariant under symmetries that keep $\|\cdot\|_\eta$ invariant. A general invariant energy function can combine $r_{ij}$ for different $i, j$ in arbitrary ways. Usually in physical systems each pair contributes an additive term in to the total energy. Assuming additivity, the energy has a form

$$\mathscr{L}(X) = \sum_{ij} f_{ij}(r_{ij}) \tag{17}$$

where $f_{ij}(z) = f_{ji}(z)$ (symmetric under $i \leftrightarrow j$). For example, when particle $i$ has electric charge $q_i$, the Coulomb potential between $i, j$ can be written as in equation 17 using $f_{ij}(z) = kq_i q_j / z$. Similarly, weak van der Waals (vdW) forces in molecular systems, which are modeled as Lennard-Jones potential, are also of the form in equation 17 with

$$\text{van der Waals:} \quad f_{ij}(r_{ij}) = V_{p,q}\left(\frac{r_{ij}}{\sigma_{ij}}\right), \quad V_{p,q}(r) = \frac{1}{r^p} - \frac{1}{r^q}. \tag{18}$$

Here $\sigma_{ij} = a_i + a_j$, where $a_i$ is the vdW radius of particle $i$, and vdW uses $p = 12, q = 6$. Next, we show that the Hessian of equation 17 has an important property which aids in finding its slow modes.

## D.1   Hessian of invariant potentials

The Hessian of potentials of the form equation 17 has the special property that it is the graph Laplacian of a weighted graph which depends on $X$, as we show now (see appendix E for details). This will play a crucial role in our argument about canonical slow modes.

**Hessian as a graph Laplacian.**   Let $\partial_i \equiv \partial/\partial X_i$ and let $\hat{r} = \eta r / r$ be the dual unit vector of $\boldsymbol{r}$. First, observe that $\partial_i r_{jk} = \hat{r}_{jk}(\delta_{ij} - \delta_{ik})$ where $\hat{r}_{jk}$ is the unit vector of $\boldsymbol{r}_{jk}$ and $\delta_{ij}$ is the Kronecker delta (1 if $i = j$, 0 otherwise). Let $\mathrm{Hes}[g]$ denote the Hessian of a function $g$. We find that (app. E)

$$\mathrm{Hes}[\mathscr{L}](X)_{ij} = \partial_i \partial_j \mathscr{L}(X) = \sum_k (\delta_{ij} - \delta_{jk})\boldsymbol{H}_{ik}(X) \tag{19}$$

where $\boldsymbol{H}_{ik}(X) = \mathrm{Hes}[f_{ik}](r_{ik})$ and given by

$$\boldsymbol{H}_{ik}(X) = \left[\left(f_{ik}''(v) - \frac{f_{ik}'(v)}{v}\right)\hat{v} \otimes \hat{v} + \frac{f_{ik}'(v)}{v}\eta\right]_{\boldsymbol{v}=r_{ik}} \tag{20}$$

Note that $\boldsymbol{H}$ has four indices, with components $\boldsymbol{H}_{ij}^{\mu\nu}$, having two particle indices $i, j$ and two spatial indices $\mu, \nu$. Recall the Laplacian of an undirected graph with adjacency matrix $A$ is defined as $L = \mathrm{Lap}(A) = D - A$, where $D$ is the degree matrix with elements $D_{ij} = \delta_{ij} \sum_k A_{ik}$. The components of Laplacian can also be written as $L_{ij} = \sum_k A_{ik}(\delta_{ij} - \delta_{jk})$. Thus, we see that the Hessian of $\mathscr{L}$ is indeed the Laplacian of $\boldsymbol{H}$

$$\mathrm{Hes}[\mathscr{L}](X)_{ij} = \sum_k (\delta_{ij} - \delta_{jk})\boldsymbol{H}_{ik} = \mathrm{Lap}(\boldsymbol{H})_{ij} \tag{21}$$

where for every pair of spatial indices the Hessian is a Laplacian over particle indices. The Hessian being Laplacian has an important effect on its null eigenvectors. To show this we make use of the incidence matrix.

## D.2 Canonical backbone for the Hessian

As the Hessian depends on $X$, it is not clear whether slow modes found at a given $X$ would be applicable to other $X$. We need some guarantee that a set of modes exist which are approximately slow modes for the Hessian at a range of different $X$. We could use multiple perturbed configurations $X + \delta X$ with random $\delta X \sim \mathcal{N}(0, T)$ to get an ensemble of Hessians $\mathcal{H} = \{H(X + \delta X)\}$ and find the overlap of the slow modes of the Hessians in $\mathcal{H}$. However, this is expensive, roughly $O(mkn^2)$ for $m = |\mathcal{H}|$ and $k$ slow modes. We cannot recompute the Hessian slow modes often. We also want a method which is more efficient than quasi-Newton methods such as BFGS. Our solution is to find a backbone for the sampled Hessians whose slow modes are guaranteed to be approximate slow modes of the actual Hessians. The key observation is that the Hessian in equation 21 is a Laplacian of a weighted graph. We show that the slow modes of weighted Laplacians overlap significantly with their unweighted counterparts.

We want to extract a set of slow modes from the sampled Hessians $H(X')$. We then compute a backbone from these Hessians of the form

$$\text{Backbone:} \quad \mathbf{H}_{ij} = \sum_{X' \in \mathbf{Sample}(X)} \|H_{ij}(X')\|^2 \qquad (22)$$

Here $i, j \in \mathbb{Z}_n$ are the particle indices and the Frobenius norm $\|H_{ij}\|^2 = \sum_{\mu, \nu} (H_{ij}^{\mu\nu})^2$ sums over the feature indices (note that $X_i^\mu$ has a particle index $i$ and a feature index $\mu \in \{1, \ldots d\}$). Then, we extract the slow modes of the backbone, by doing a spectral expansion $\mathbf{H} = \sum_i \lambda_i \psi_i \psi_i^T$ and picking $\psi_i$ with $|\lambda_i| < \varepsilon^2 \max_j[\lambda_j]$, for some small $\varepsilon < 1$.

The intuition behind equation 22 is to identify the components in the sampled Hessians which have consistently high magnitudes. If we had taken a simple mean we could get very small values, because the components can fluctuate randomly. Also, if we had taken the variance instead of the norm, we would get zero for quadratic $\mathcal{L}$, where $H$ is constant and has no variance. However, these intuitions do not show that there would be any connection between the modes of the backbone $\mathbf{H}$ and the actual Hessians $H(X')$. Importantly, entries in $H(X')$ have signs, which affects the spectrum, whereas all entries in $\mathbf{H}$ are positive. So why should the spectra of $H$ and $\mathbf{H}$ be related? This is where the structure of $\mathcal{L}$ comes into play. Indeed, as we show below, for many physical $\mathcal{L}$, the slow modes of the backbone $\mathbf{H}$ approximate the slow modes of sampled $H(X')$ up to $O(\varepsilon^2)$ errors.

**Definition D.1** (weighted graph). *Let $\hat{\mathcal{G}} = (\mathcal{V}, \mathcal{E})$ be a graph with vertices $\mathcal{V} = \mathbb{Z}_n$, edges $\mathcal{E} \subseteq \mathcal{V} \times \mathcal{V}$. Let $\hat{A} \in \mathbb{R}^{n \times n}$ denote the adjacency matrix $\hat{A}_{ij} = 1$ if $(i, j) \in \mathcal{E}$ and 0 otherwise. We denote a weighted graph as $\mathcal{G} = (\mathcal{V}, \mathcal{E}, \mathcal{W})$ where $\mathcal{W} : \mathcal{E} \to \mathbb{R}$ are the weights of the edges. Let $A$ denote the adjacency matrix of $\mathcal{G}$, where $A_{ij} = \mathcal{W}(i, j)$ or zero if $(i, j) \notin \mathcal{E}$. The Laplacian $L = \mathrm{Lap}(A)$ of an undirected weighted graph is defined analogous to the unweighted graph as $L = D - A$ with degree matrix elements $D_{ij} = \delta_{ik} \sum_k A_{ik}$.*

**Definition D.2** (Slow manifold). *Let $L$ be a graph Laplacian (undirected, weighted or unweighted), with spectral expansion $L = \sum_{i=1}^n \lambda_i \psi_i \psi_i^T$. Let $\varepsilon \ll 1$ and $\lambda_{\max} = \max\{\lambda_i\}$ be the largest eigenvalue of $L$. We define the slow manifold as*

$$\mathbf{Slow}_\varepsilon[L] = \mathrm{Span}\{\psi_i \,|\, |\lambda_i| < \varepsilon^2 \lambda_{\max}\} \qquad (23)$$

**Theorem D.1** (Slow modes of weighted Laplacians). *Let $A$ be the adjacency matrix of a weighted graph and $\hat{A}$ be its unweighted counterpart. Let $L = \mathrm{Lap}(A)$ and $\hat{L} = \mathrm{Lap}(\hat{A})$. Then $\mathbf{Slow}_\varepsilon[L]$ overlaps with $\mathbf{Slow}_\varepsilon[\hat{L}]$ up to $O(\varepsilon^2)$ corrections from the rest of the modes.*

To prove this we will make use of the incidence matrix representation of the Laplacian.

**Definition D.3** (Incidence matrix). *Given a weighted graph $\mathcal{G} = (\mathcal{V}, \mathcal{E}, \mathcal{W})$, define its incidence matrix as $C : \mathcal{V} \times \mathcal{E} \to \{\pm 1\}$, where for any edge $e = (i \to j) \in \mathcal{E}$, $C_{i,e} = -1$ and $C_{j,e} = 1$, and zero for other components.*

**Lemma D.2** (Laplacian in terms of the incidence matrix). *Let $w = \mathrm{vec}(\mathcal{W}(\mathcal{E}))$ be the vector of all weights indexed in the same order as the columns of $C$, with $w_e = A_{ij}$, for $e = (i, j)$ and let $W$ be a diagonal matrix with $w$ on its diagonal. Then, the Laplacian $L = \mathrm{Lap}(A)$ can be written as $L = \frac{1}{2} CWC^T$ (proof in app. E.1).*

Because $\mathcal{G}$ and $\hat{\mathcal{G}}$ share the same vertices and edges, their incidence matrix $C$ is the same. From Lemma D.2, $L = \frac{1}{2} CWC^T$ and $\hat{L} = \frac{1}{2} CC^T$ as $\hat{\mathcal{G}}$ is unweighted. Using SVD, $C = USV^T$ and

defining $R = US/\sqrt{2}$ and $Q = V^T W V$, we have

$$\hat{L} = RR^T \qquad\qquad\qquad L = RQR^T. \qquad (24)$$

Note that for a random configuration $X$ the edge weights $W$ will be random, as they arising from derivatives of $f_{ij}(r_{ij})$ in equation 20 (unless $f_{ij}$ is quadratic which makes $W$ constant). Therefore, we will assume $Q$ has a uniform Gaussian distribution. Assuming $W$ is also Gaussian, the spectrum of such a $Q = V^T W V$ is somewhere between the distribution of $W$ (for sparse graphs with $|\mathcal{E}| \sim O(|\mathcal{V}|)$) and a Wigner Semi-circle (for dense graphs with $|\mathcal{E}| \sim O(|\mathcal{V}|^2)$). See appendix E.2 for more discussion. We also assume $Q$ has no particular block structure and that the spectrum of any diagonal block of $Q$ should also follows a distribution similar to all of $Q$.

**Slow subspace.** We now sketch the proof for Theorem D.1. For details, refer to appendix E.4. From the SVD, $C = USV^T$, the slow subspace is

$$\mathbf{Slow}_\varepsilon[\hat{L}] = \left\{ i \big| S_{ii} < \varepsilon \max[S] \right\} \qquad (25)$$

Normalize $\hat{S} = S/\max[S]$ and make them all positive (e.g. absorb their sign into $U$). For some $\varepsilon < 1$ sort the SV such that $\hat{S} = \mathrm{diag}(S_\varepsilon, S_1)$ where the diagonal matrices $S_\varepsilon < \varepsilon$ and $S_1 \geq \varepsilon$. Now, the problem of finding $\mathbf{Slow}_\varepsilon[L]$ becomes finding eigenvectors of the matrix $\hat{M} = \hat{S} Q \hat{S}^T$ with eigenvalues $O(\varepsilon^2)$. Using $S_\varepsilon \sim O(\varepsilon)$ and $S_1 \sim O(1)$, we can pull factors of $\varepsilon$ out from $\hat{M}$ and write it as

$$\hat{M} = M_0 + \hat{\varepsilon}\delta M, \qquad M_0 = \begin{pmatrix} \hat{\varepsilon}^2 \hat{A} & 0 \\ 0 & C \end{pmatrix}, \qquad \delta M = \begin{pmatrix} 0 & \hat{B} \\ \hat{B}^T & 0 \end{pmatrix}. \qquad (26)$$

where $\hat{\varepsilon}^2 \equiv \varepsilon^2 \sqrt{n_A/n_C}$ is rescaled so that the random matrices $\hat{A} \in \mathbb{R}^{n_A \times n_A}$ and $C \in \mathbb{R}^{n_C \times n_C}$ have a similar range of eigenvalues. Next, using a perturbative ansatz for eigenvectors $\psi' = \psi + \hat{\varepsilon}\delta\psi$ and eigenvalues $\lambda' = \lambda + \hat{\varepsilon}\delta\lambda$, we solve $\hat{M}\psi' = \lambda'\psi'$ up to $O(\hat{\varepsilon}^2)$ corrections.

To find slow modes for $L$ we start from $\psi \in \mathbf{Slow}_\varepsilon[\hat{L}]$. Specifically, we start with an eigenvector $\psi_A$ of $\hat{A}$ and concatenate it with zeros to get $\psi = (\psi_A, 0)$. We have $M_0 \psi = \lambda\psi$ with $\lambda = \hat{\varepsilon}^2 \lambda_A$. Using first-order perturbation theory, we find the corrections $\delta\lambda$ to the eigenvalues and eigenvectors to be

$$\delta\lambda = \psi^T \delta M \psi = 0, \qquad \delta\psi = -(M_0 - \lambda)^{-1}\delta M \psi = \begin{pmatrix} 0 \\ (C - \lambda)^{-1}\hat{B}^T\psi_A \end{pmatrix}. \qquad (27)$$

Putting all together we find the slow eigenvector $\psi' = \psi + \hat{\varepsilon}\delta\psi$ up to order $O(\varepsilon^2)$ to be

$$\mathbf{Slow}_\varepsilon[L] \ni \psi' = \begin{pmatrix} \psi_A \\ \hat{\varepsilon}(C - \hat{\varepsilon}^2\lambda_A)^{-1}\hat{B}^T\psi_A \end{pmatrix}, \qquad \hat{M}\psi' = \hat{\varepsilon}^2\lambda\psi' + O(\hat{\varepsilon}^2) = O(\hat{\varepsilon}^2) \qquad (28)$$

meaning to first order in $\hat{\varepsilon}$ the corrections to eigenvalues of slow modes vanishes. This is desired because the slow mode eigenvalues are $O(\hat{\varepsilon}^2)$. We also observe that slow modes of $L$ are mostly confined to $\mathbf{Slow}_\varepsilon[\hat{L}]$ and only get $O(\varepsilon)$ contributions from the fast subspace of $\hat{L}$.

As a side, it follows that all weighted graphs share the null space of the unweighted Laplacian.

**Proposition D.3** (Shared null space). *Let $\mathbf{Null}[M] = \mathrm{Span}\{v | v \in \mathbb{R}^n, Mv = 0\}$ denote the null space of a matrix $M \in \mathbb{R}^{n \times n}$. The null space of the Laplacian $\hat{L}$ (unweighted) is contained in the null space of Laplacian $L$ (weighted), meaning $\mathbf{Null}[\hat{L}] \subseteq \mathbf{Null}[L]$.*

**Lemma D.4.** $\mathbf{Null}[\hat{L}] = \mathbf{Null}[R^T]$

*Proof.* $\forall v \in \mathbf{Null}[\hat{L}], 0 = v^T \hat{L} v = \|R^T v\|^2$ and $\forall v \in \mathbf{Null}[R^T], \hat{L}v = RR^T v = 0$. □

*Proof of proposition D.3.* $\forall v \in \mathbf{Null}[\hat{L}], Lv = RQR^T v = 0$ hence, $\mathbf{Null}[\hat{L}] \subseteq \mathbf{Null}[L]$. □

Note that $\mathbf{Null}[\hat{L}]$ and $\subseteq \mathbf{Null}[L]$ are not necessarily the same because weights can be zero, which could make the null space of the weighted graph larger than the unweighted one. Next, we present our method for coarse-graining using a set of canonical slow modes.

## E  Invariant additive dyadic potentials

We want to Compute the Hessian of equation 17, $\mathscr{L}(X) = \sum_{ij}(r_{ij})$. Let $\hat{r} = \eta r/r$ be the dual unit vector of $r$. First, note that

$$
\begin{aligned}
\partial_i r_{jk} \equiv \frac{\partial r_{jk}}{\partial X_i} &= \partial_i \sqrt{\|X_j - X_k\|_\eta} \\
&= \eta \frac{r_{jk}}{r_{jk}} (\delta_{ij} - \delta_{ik}) = \hat{r}_{jk}(\delta_{ij} - \delta_{ik})
\end{aligned}
\tag{29}
$$

Then the gradient becomes

$$
\begin{aligned}
\partial_i \mathscr{L}(X) &= \sum_{j,k} f'_{jk}(r_{jk}) \frac{\partial r_{jk}}{\partial x_i} \\
&= \sum_{j,k} f'_{jk}(r_{jk}) \eta \hat{r}_{jk}(\delta_{ij} - \delta_{ik}) \\
&= 2 \sum_j f'_{ij}(r_{ij}) \eta \hat{r}_{ij}.
\end{aligned}
\tag{30}
$$

where we used $\hat{r}_{jk} = -\hat{r}_{kj}$ to show both terms in $(\delta_{ij} - \delta_{ik})$ yield the same output. Finally, the Hessian becomes

$$
\begin{aligned}
[H(X)]_{ij} = \partial_i \partial_j \mathscr{L}(X) &= 2 \partial_j \sum_k f'_{ik}(r_{ik}) \hat{r}_{ik} \\
&= 2 \sum_k [f''_{ik}(r_{ik}) \partial_j r_{ik} \otimes \hat{r}_{ik} + f'_{ik}(r_{ik}) \partial_j \hat{r}_{ik}] \\
&= 2 \sum_k \Bigg[ (\delta_{ji} - \delta_{jk}) f''_{ik}(r_{ik}) \hat{r}_{ik} \otimes \hat{r}_{ik} \\
&\quad + f'_{ik}(r_{ik}) \left( \eta \frac{\delta_{ji} - \delta_{jk}}{r_{ik}} - \frac{\hat{r}_{ik}}{r^2_{ik}} \partial_j r_{ik} \right) \Bigg] \\
&= 2 \sum_k (\delta_{ji} - \delta_{jk}) \left[ f''_{ik}(r_{ik}) \hat{r}_{ik} \otimes \hat{r}_{ik} + f'_{ik}(r_{ik}) \left( \frac{\eta}{r_{ik}} - \frac{\hat{r}_{ik}}{r^2_{ik}} \otimes \hat{r}_{ik} \right) \right] \\
&= 2 \sum_k \left[ f''_{ik}(v) \hat{v} \otimes \hat{v} + \frac{f'_{ik}(v)}{v} (\eta - \hat{v} \otimes \hat{v}) \right]_{v=r_{ik}} (\delta_{ij} - \delta_{jk}) \\
&= 2 \sum_k \left[ \left( f''_{ik}(v) - \frac{f'_{ik}(v)}{v} \right) \hat{v} \otimes \hat{v} + \frac{f'_{ik}(v)}{v} \eta \right]_{v=r_{ik}} (\delta_{ij} - \delta_{jk}) \\
&= \sum_k H_{ik}(x) (\delta_{ij} - \delta_{jk}) = \mathrm{Lap}(H)_{ij}
\end{aligned}
\tag{31}
$$

This is because the components of Laplacian can be be written

$$
\begin{aligned}
L_{ij} = \mathrm{Lap}(A)_{ij} &= (D - A)_{ij} \\
&= \delta_{ij} \sum_k A_{ik} - A_{ij} = \sum_k A_{ik}(\delta_{ij} - \delta_{jk})
\end{aligned}
\tag{32}
$$

## E.1 Incidence matrix

The Laplacian $L = D - A$ of an undirected graph with adjacency $A$ can be written as $L = CWC^T/2$ using the incidence matrix $C$ and the edge weights $W$. This can be shown as follows

$$
\begin{aligned}
[CWC^T]_{ij} &= \sum_e C_i^e W_{ee} C_j^e \\
&= \sum_{k,l} C_i^{(k \to l)} A_{kl} C_j^{(k \to l)} \\
&= \sum_{k,l} (\delta_{il} - \delta_{ik}) A_{kl} (\delta_{jl} - \delta_{jk}) \\
&= \sum_{k,l} (\delta_{il}\delta_{jl} - \delta_{ik}\delta_{jl} - \delta_{il}\delta_{jk} + \delta_{ik}\delta_{jk}) A_{kl} \\
&= 2\sum_{k,l} (\delta_{il}\delta_{jl} - \delta_{ik}\delta_{jl}) A_{kl} \\
&= 2\sum_k \delta_{ij} A_{kj} - 2A_{ij} = 2(D-A)_{ij} = 2L_{ij}
\end{aligned}
\tag{33}
$$

where we assumed $A_{kl} = A_{lk}$ (undirected graph).

So the same derivation of the backbone also holds for this case. The idea is that using the incidence matrix $C$ and edge weights $W$ (as a diagonal matrix), any Laplacian $L$ can be decomposed as $L = CWC^T$. Then, doing SVD $C = USV^T$ we have

$$
L = USV^T WVS^T U^T = UMU^T
\tag{34}
$$

Where the matrix $M = SV^T WVS^T$ has an interesting property, namely that its null space includes the null space of the unweighted Laplacian $L_0 = CC^T$. To see this note that $L_0 = USS^T S^T$, which means columns $U_i$ are the eigenvectors of $L_0$ with eigenvalues $S_i^2$. The null eigenspace of $L_0$ are the $U_i$ for which $S_i = 0$. This subspace will also be a null subspace for $L$, because that block is also zero in $M$, because $M_{ij} = \sum_c S_i V_{ik} W_{kk} V_{jk} S_j$. So, whenever $S_i = 0$ or $S_j = 0$, $M_{ij} = 0$, meaning that whole block in $M$ is zero and $MU_i = 0$ ( write it better).

**Example: power law.** Let $f(r) = r^p$. We have $f' = pr^{p-1}$ and $f'' = p(p-1)r^{p-2}$, yielding the Hessian

$$
\boldsymbol{H} = \nabla\nabla f(r) = r^{p-2}\left[\left(p^2 - 2p\right)\hat{r} \otimes \hat{r} + p\eta\right]
\tag{35}
$$

$$
B_{ik} = A_{ik} r_{ik}^{p-2}\left[\left(p^2 - 2p\right)\hat{r}_{ik} \otimes \hat{r}_{ik} + p\eta\right]
\tag{36}
$$

**Example: Lennard-Jones.** This potential has the form

$$
f(r) = 4\varepsilon\left[\left(\frac{\sigma}{r}\right)^p - \left(\frac{\sigma}{r}\right)^q\right]
\tag{37}
$$

where for classic van-der Waals potential $p = 2q = 12$. The Hessian for this potential is given by

$$
\boldsymbol{H}(r) = \nabla\nabla f(r) = \varepsilon\left[\left(\frac{\sigma}{r}\right)^{p+2}\left[\left(p^2 + 2p\right)\hat{r} \otimes \hat{r} - p\eta\right] - \left(\frac{\sigma}{r}\right)^{q+2}\left[\left(q^2 + 2q\right)\hat{r} \otimes \hat{r} - q\eta\right]\right]
\tag{38}
$$

and $B_{ik} = A_{ik}\boldsymbol{H}(r_{ik})$

## E.2 Structure and spectrum of of $Q = V^T WV$

To consider only the relevant subspace of SVD, we have $U, S \in \mathbb{R}^{n \times n}$, and $V \in \mathbb{R}^{m \times n}$, with $n = |\mathcal{V}|$ and $m = |\mathcal{E}|$. For a connected undirected graph $m \geq 2(n-1)$ and $V$ is full rank ($V^T V = I_n$). Note the edge weights $W$ come from the forces $f_{ij}(r_{ij})$ in equation 20, which for an arbitrary $X$ will be random. Assuming a Gaussian distribution $W_{ee} \sim \mathcal{N}(0, \sigma)$ for all edges $e$, the matrix $Q$

will also have random Gaussian entries. When $m = n$, $V$ defines the eigenbasis of $Q$ and $W_{ee}$ are the eigenvalues of $Q$. Similarly, in sparse graphs, where $m \sim O(n)$, $V$ is approximately the eigenbasis and the spectrum of $Q$ should have a distribution similar to $W_{ee}$. For dense graphs, where $m \sim O(n^2)$, every entry of $Q$ will involve a weighted sum over multiple $W_{ee}$. Then, from central limit theorem, entries of $Q$ will asymptotically have a Gaussian distribution. From random matrix theory, we know that such $Q$ will have a spectrum which follows the Wigner-semi-circle law. In both cases (sparse and dense graphs) the spectrum of $Q$ has a finite variance and sits somewhere between a Gaussian and a semi-circle.

### E.3 Generalization to nonzero but small SV

We want to know how much the slow modes of weighted and unweighted graphs to overlaps. With the spectral expansion $\hat{L} = \sum_i \lambda_i \psi_i \psi_i^T$ Define the slow subspace as in equation 23

$$\mathbf{Slow}_\varepsilon[\hat{L}] = \mathrm{Span}\{\psi_i \big| |\lambda_i| < \varepsilon^2 \lambda_{\max}(\hat{L})\} \tag{39}$$

where $\lambda_{\max}(\hat{L}) = \max\{\lambda_i\} = \max_\psi[\psi^T L \psi / \|\psi\|^2]$ is the largest eigenvalue of $L$ and $\varepsilon \ll 1$. In terms of the singular values (SV) of the incidence matrix $C = USV^T$, the slow subspace becomes

$$\mathbf{Slow}_\varepsilon[\hat{L}] = \{i \big| S_{ii} < \varepsilon \max[S]\} \tag{40}$$

We will show that the slow modes in weighted $L = CWC^T$ are perturbations to the slow modes of $\hat{L}$. Define

$$M = SV^T W V S^T = SQS^T \tag{41}$$

Normalize $\hat{S} = S / \max[S]$. Break the space down to the slow and fast subspaces, based on whether $\hat{S}_{ii} < \varepsilon$ or not. First, since $L$ is positive semi-definite, we can make all $S_{ii} \geq 0$. Let $\hat{S} = S / \max S$. We sort the dimensions in $\hat{S}$ to have the small SVs appear first. Denote the block in $\hat{S}$ where $S_{ii} < \varepsilon$ by $S_\varepsilon$. We have

$$\hat{S}^2 = \begin{pmatrix} S_\varepsilon^2 & 0 \\ 0 & S_1^2 \end{pmatrix} < \begin{pmatrix} \varepsilon^2 & 0 \\ 0 & 1 \end{pmatrix} \tag{42}$$

We know the null space of $\hat{L}$, where $S_{ii} = 0$, is shared with $L$. First, we remove the null space from $L$ and $\hat{L}$, calling the remainder $L_0$ and $\hat{L}_0$ and the remaining SVs $\hat{S}$. Then in this remainder subspace we need to find parts which are $O(\varepsilon)$. We sort the dimensions in $\hat{S}$ to have the small SVs appear first. We denote the block in $\hat{S}$ where $S_{ii}^2 < \varepsilon \max[S^2]$ by $S_\varepsilon$. We have

$$M = \max[S]^2 \hat{S} Q \hat{S}^T = \begin{pmatrix} S_\varepsilon Q_{\varepsilon\varepsilon} S_\varepsilon & S_\varepsilon Q_{\varepsilon 1} S_1 \\ S_1 Q_{\varepsilon 1}^T S_\varepsilon & S_1 Q_{11} S_1 \end{pmatrix} = \begin{pmatrix} M_{\varepsilon\varepsilon} & M_{\varepsilon 1} \\ M_{\varepsilon 1}^T & M_{11} \end{pmatrix} \tag{43}$$

Because $S_\varepsilon$ is $O(\varepsilon)$ and $S_1$ is $O(1)$, we will factor out the factors of $\varepsilon$ from blocks in $M$ and write

$$M = \max[S]^2 \begin{pmatrix} \varepsilon^2 A & \varepsilon B \\ \varepsilon B^T & C \end{pmatrix} \tag{44}$$

Here $A$ and $C$ are random matrices built from their corresponding blocks in $Q$ and sandwiched between $S_\varepsilon / \varepsilon$ (for $A$), and $S_1$ (for $C$), which have $O(1)$ values. The spectrum of $Q$ has a distribution between a Gaussian with mean zero and a Wigner semi-circle, also centered around zero. We expect spectra of $A$ and $C$ to be similar to $Q$. Denote the spectral expansion of $Q$ as

$$Q = \Psi \Lambda \Psi^T, \qquad \Lambda = \mathrm{diag}(\lambda_i)_{i=1}^n, \qquad \Psi = [\psi_i]_{i=1}^n. \tag{45}$$

This is because when $Q_{ij} \sim \mathcal{N}(0, \sigma)$ we have (ignoring Bessel's correction for $k \gg 1$).

$$\sigma^2 = \mathrm{Var}(Q_{ij}) \approx \frac{1}{n} \|Q\|^2 = \frac{1}{n} \sum_i \lambda_i^2 = \mathrm{Var}(\Lambda) \tag{46}$$

where we assumed $\mathrm{Tr}\{Q\}/n \approx \mathrm{mean}(Q) = 0$. Since a block $Q_k$ of size $k$ is $k^2$ entries sampled from the same distribution as $Q$, we expect

$$\frac{\|Q_k\|^2}{k^2} \approx \frac{\|Q_l\|^2}{l^2} \quad \Rightarrow \frac{1}{k} \mathrm{Var}(Q_k) \approx \frac{1}{l} \mathrm{Var}(Q_l) \tag{47}$$

Thus, rescaling $A \in \mathbb{R}^{n_A \times n_A}$ and $C \in \mathbb{R}^{n_C \times n_C}$ we get

$$\hat{A} = \frac{A}{\sqrt{n_A}}, \quad \hat{C} = \frac{C}{\sqrt{n_C}}, \quad \mathrm{Var}(\hat{A}) \approx \mathrm{Var}(\hat{C}) \tag{48}$$

## E.4 Approximate slow modes of $L$

If $M$ did not have the off-diagonal blocks $B$, then $\mathbf{Slow}_\varepsilon[L]$ and $\mathbf{Slow}_\varepsilon[\hat{L}]$ would coincide, as the $S_\varepsilon$ block and the $S_1$ block would not mix when $B = 0$. Define $M_0$ as the block matrix of $M$ with $B = 0$.

$$M_0 \equiv \begin{pmatrix} \varepsilon^2 A & 0 \\ 0 & C \end{pmatrix} \tag{49}$$

Using spectral expansions

$$A = \Psi_A \Lambda_A \Psi_A^T, \quad C = \Psi_C \Lambda_C \Psi_C^T \tag{50}$$

the eigenvectors of $M_0$ consist of

$$M_0 \begin{pmatrix} \psi_{Ai} \\ 0 \end{pmatrix} = \varepsilon^2 \lambda_{Ai} \begin{pmatrix} \psi_{Ai} \\ 0 \end{pmatrix}, \qquad M_0 \begin{pmatrix} 0 \\ \psi_{Ci} \end{pmatrix} = \lambda_{Ci} \begin{pmatrix} 0 \\ \psi_{Ci} \end{pmatrix}. \tag{51}$$

Since we are looking for slow modes, we must also consider the magnitudes of $\lambda_{Ai}$ and $\lambda_{Ci}$. Since $A$ and $C$ entries are random samples from $Q$, we expect them to have a semi-circle or Gaussian distribution similar to $Q$. Thus, we can use the variances of eigenvalues of $A$ and $C$ as a proxy for the how the magnitudes of $\lambda_{Ai}$ and $\lambda_{Ci}$ compare. From equation 48 we have

$$\frac{1}{n_A}\mathbb{E}[\Lambda_A^2] \approx \frac{1}{n_A}\mathrm{Var}(A) \approx \frac{1}{n_C}\mathbb{E}[\Lambda_C^2] \tag{52}$$

Based on this we define a rescaled $\hat{\varepsilon}$ such that $\varepsilon^2 \lambda_{Ai}$ still has a smaller magnitude than $\lambda_{Ci}$ on average, meaning we want

$$\varepsilon^4 \mathbb{E}[\Lambda_A^2] < \mathbb{E}[\Lambda_C^2] \qquad \Rightarrow \qquad \varepsilon^4 n_A < n_C \qquad \Rightarrow \qquad \hat{\varepsilon}^2 \equiv \varepsilon^2 \sqrt{\frac{n_A}{n_C}} < 1 \tag{53}$$

We choose $\varepsilon$ such that the condition in equation 53 is satisfied. We can express $M$ in terms of $\hat{\varepsilon}$ by rescaling $A$ and $B$ to $\hat{\varepsilon}^2 \hat{A} = \varepsilon^2 A$ and $\hat{\varepsilon}\hat{B} = \varepsilon B$. Now eigenvalues of $\hat{A}$ have the same variance as eigenvalues of $C$. For brevity, denote $\hat{M} = \max[S]^2 M$. We have

$$\hat{M} = \begin{pmatrix} \hat{\varepsilon}^2 \hat{A} & \hat{\varepsilon}\hat{B} \\ \hat{\varepsilon}\hat{B}^T & C \end{pmatrix}. \tag{54}$$

To find how slow modes of $\hat{M} = SQS^T / \max[S]^2$ differ from slow modes of $SS^T$, we break $\hat{M}$ into a block diagonal part and an $O(\hat{\varepsilon})$ off-diagonal perturbation

$$\hat{M} = M_0 + \hat{\varepsilon}\delta M, \qquad M_0 = \begin{pmatrix} \hat{\varepsilon}^2 \hat{A} & 0 \\ 0 & C \end{pmatrix} \qquad \delta M = \begin{pmatrix} 0 & \hat{B} \\ \hat{B}^T & 0 \end{pmatrix}. \tag{55}$$

As in equation 51, eigenvectors of $A = \sqrt{n_C/n_A}\hat{A}$ and $C$ are eigenvectors of $M_0$. Now we want to find eigenvectors of $\hat{M}$ with small $O(\varepsilon^2)$ eigenvalues up to order $\hat{\varepsilon}$ corrections by treating $\delta M$ as a perturbation.

$$(M_0 + \hat{\varepsilon}\delta M)(\psi + \hat{\varepsilon}\delta\psi) = (\lambda + \hat{\varepsilon}\delta\lambda)(\psi + \hat{\varepsilon}\delta\psi)$$
$$M_0\psi + \hat{\varepsilon}(\delta M\psi + M_0\delta\psi) + O(\hat{\varepsilon}^2) = \lambda\psi + \hat{\varepsilon}(\delta\lambda\psi + \lambda\delta\psi) + O(\hat{\varepsilon}^2)$$
$$\Rightarrow \delta M\psi + M_0\delta\psi = \delta\lambda\psi + \lambda\delta\psi \tag{56}$$

We only need the components of $\delta\psi$ orthogonal to $\psi$, so we can assume $\delta\psi^T\psi = 0$. From this we have

$$\delta\lambda = \psi^T \delta M\psi + \psi^T M_0\delta\psi = \psi^T \delta M\psi, \tag{57}$$

where we used $\psi^T M_0\delta\psi = \lambda\psi^T\delta\psi = 0$. Plugging equation 57 into equation 56 we can solve for $\delta\psi$ by inverting the matrices

$$(M_0 - \lambda)\delta\psi = (\delta\lambda - \delta M)\psi$$
$$\Rightarrow \delta\psi = (M_0 - \lambda + i\eta)^{-1}(\delta\lambda - \delta M)\psi \tag{58}$$

where we added a small $\eta$ to make the matrix $M_0 - \lambda$ invertible, as $\lambda$ is one of its eigenvalues.

To find slow modes, we start from slow modes of $M_0$ which are in the $A$ subspace. Let $\psi_A$ be an eigenvector of $A$ with $\hat{A}\psi_A = \lambda_A\psi_A$. Concatenating $\psi_A$ with zeros in the $C$ subspace we have

$$\psi = \begin{pmatrix} \psi_A \\ 0 \end{pmatrix}, \qquad\qquad M_0\psi = \hat{\varepsilon}^2\lambda_A\psi. \tag{59}$$

Using this $\psi$ to compute $\delta\lambda$ in equation 57 we have

$$\delta\lambda = \psi^T\delta M\psi = \begin{pmatrix} \psi_A^T & 0 \end{pmatrix} \begin{pmatrix} 0 \\ \hat{B}^T\psi_A \end{pmatrix} = 0 \tag{60}$$

meaning to first order in $\hat{\varepsilon}$ the corrections to eigenvalues of slow modes vanishes. This is desired because the slow mode eigenvalues are $O(\hat{\varepsilon}^2)$ and we find that with this $\psi$ ansatz the corrections it will get are also at least $O(\hat{\varepsilon}^2)$. Next, we compute the corrections $\delta\psi$ to the eigenvectors. Plugging $\psi$ into equation 58 with $\lambda = \hat{\varepsilon}^2\lambda_A$ and $\delta\lambda = 0$ we have

$$(M_0 - \lambda + i\eta)^{-1} = \begin{pmatrix} (\hat{\varepsilon}^2\hat{A} - \lambda + i\eta)^{-1} & 0 \\ 0 & (C - \lambda)^{-1} \end{pmatrix}$$
$$\delta\psi = -(M_0 - \lambda + i\eta)^{-1}\delta M\psi$$
$$= \begin{pmatrix} 0 \\ (C - \lambda)^{-1}\hat{B}^T\psi_A \end{pmatrix} \tag{61}$$

where we dropped $i\eta$ in the lower block because $\hat{\varepsilon}^2\lambda_A$ is unlikely to be also an eigenvalue of $C$, as $A$ and $C$ are random matrices.

Using the relation $\hat{\varepsilon}\hat{B} = \varepsilon B$ with the original $\varepsilon$ and putting all together we find the eigenvector $\psi' = \psi + \hat{\varepsilon}\delta\psi$ up to order $O(\varepsilon^2)$ to be

$$\psi' = \begin{pmatrix} \psi_A \\ \hat{\varepsilon}(C - \hat{\varepsilon}^2\lambda_A)^{-1}\hat{B}^T \end{pmatrix} \tag{62}$$
$$\hat{M}\psi' = \hat{\varepsilon}^2\lambda\psi' + O(\hat{\varepsilon}^2) = O(\hat{\varepsilon}^2) \tag{63}$$

