# OpenReview forum: "Neural Network Reparametrization for Accelerated Optimization in Molecular Simulations"
_NeurIPS.cc/2024/Conference — NeurIPS 2024 poster_

### Official Review · Reviewer_qFP8 · 2024-06-26

**Soundness:** 3
**Presentation:** 2
**Contribution:** 3
**Rating:** 6
**Confidence:** 4

**Summary:**

The paper proposes a novel neural network reparameterization approach which provides a flexible alternative to traditional coarse-graining methods for molecular simulations. Unlike CG methods that strictly reduce degrees of freedom, the proposed model can dynamically adjust system complexity, potentially increasing it to simplify optimization. Fitting a neural network which maps the reduced space to the fine-grained space, the approach maintains continuous access to fine-grained modes and eliminates the need for force-matching, thereby enhancing both the efficiency and accuracy of energy minimization. The framework incorporates arbitrary neural networks like GNNs, to perform reparametrization, achieving significant improvements in molecular simulations by optimizing energy minimization and convergence speeds.

**Strengths:**

1. The neural network reparameterization approach gets rid of the force-matching and the non-unique back-mapping in traditional CG methods and shows promising potential in enhancing molecular simulations.

1. The Hessian matrix provides a precise mathematical framework to identify slow modes that correspond to the most stable and significant collective motions in the system. Such an innovative strategy does not need to use an encoder-decoder for FG-CG mapping.

1. Dynamically adjusting the effective DOF allows the model to increase complexity when needed, capturing essential details without a fixed reduction in resolution, thus resulting in more accurate representations of the system. By focusing on slow modes, the method can more effectively explore the energy landscape, avoiding local minima and achieving better convergence to global minima.

1. The authors provide solid mathematical derivation to support their proposed approach.

**Weaknesses:**

1. The manuscript needs improvement: it is difficult to follow the paper as the contents are not well-organized and some crucial details are missing.
    - The introduction and background could be reorganized. It's better to first introduce the key challenge and discuss the approach and key contributions at a high level, while the details of CG and the proposed approach could be moved to Methods. This can also remove the overlapping between the Introduction and Background.
    - Due to the lack of a high-level introduction to the overall workflow (usually the last paragraph and Figure 1), the discussion on NN reparameterization and Hessian of potential energies is confusing at first glance because readers are not clear about their roles in the big picture. Similarly, the subsections in the Hessian part propose many theorems and corollaries but do not explicitly address their contribution to the proposed workflow.
    - The Experiment part should stress what the experiment settings are, where we can find the results, and what conclusions we can draw. The information is not clearly stressed even though I can find some of it in the paragraphs.
    - Some crucial information is missing, for example, the GNN training details and the loss function. Without such information, it's hard to assess the soundness of the method.
    - Figure captions also needs improvement. For example, the caption of Figure 1 consists of many short sentences, but the key takeaway are not strengthened.

1. The major contribution on machine learning of this paper is the use of a neural network to map slow modes to FG configurations. However, this neural network model is not novel and does not introduce new ML techniques or architectures. Therefore, I'm afraid that the work may not be fully aligned with the scope of the conference. Additionally, it is unclear whether a neural network is necessary for this mapping, as other simpler or more traditional methods might suffice for this task. A more thorough justification for the use of a neural network over other potential approaches is needed.

1. The manuscript involves biology background knowledge especially in the experiments. However, they are not well explained, making the experimental session more difficult to follow.

**Questions:**

1. The proposed approach has an edge over full-atom simulations and CG methods. Meanwhile, various approaches such as metadynamics with collective variables are commonly used to learn and simulate the dynamics of the system in a reduced space. Have the authors compared the proposed neural network reparametrization method with those approaches?

1. While slow modes represent directions in the configuration space along which the energy landscape is relatively flat so that they capture low-frequency motions, it is not necessarily clear that these are the most relevant directions for the system's evolution that we care about. In other words, the slow modes do not necessarily include more "science". Thus, I'm wondering whether relying solely on slow modes may overlook critical aspects of the system's behavior.

1. Is the reparameterized neural network generalizable to different molecular systems or molecular structures under different conditions (e.g. temperature, pressure)?

1. Empirical potentials are used in this work. However, empirical potentials cannot achieve high accuracy. Have the authors tried machine learning potentials for MD simulations?

**Limitations:**

The authors discussed the limitation of lack of experiments for comparison against traditional CG methods. No potential negative societal impact is involved.

---

> ### Author Rebuttal · Authors · 2024-08-05
>
> Thank you for your valuable feedback. We agree that the manuscript requires substantial revisions for clarity and flow. Below, we aim to address each of your concerns, and to resolve the criticisms thoroughly.
>
> ## Weaknesses
> __1.__ We agree that intro and background will benefit from significant reorganization.
> 1. Our revised intro will start with an overview of challenges in scientific simulations, such as the proliferation of saddle points and local minima leading to suboptimal results. While conventional dim reduction methods like CG offer partial solutions, they encounter issues like back-mapping and force-matching. Instead, we propose an innovative approach using an overparametrized neural ansatz. We demonstrate that CG reparam or a well-designed GNN ansatz, incorporating Hessian slow modes, achieve significantly lower energy states compared to direct optimization.
> 2. We are making a new Fig. 1 (see attached pdf) to outline the methodology.
> 3. In Sec. 2 on the Hessian, we'll clarify the motivation for using slow modes, rooted in the difference in fast vs slow mode evolution rate, which causes slow convergence at saddle points. Our goal is to adapt the optimization process to grant direct access to slow modes, hoping that it helps escape such saddle points. However, this approach faces challenges: 1) Changes in the Hessian may alter the slow modes during optimization. 2) The need to modify the optimization to favor slow modes. We address these by showing the stability and robustness of slow modes and by proposing linear CG and GNN reparam. Our experiments show superior efficacy of the GNN approach.
> 4. We add GNN parameter details in a table in the appendix. For the experiments in Fig 1, the GNN hidden dims are [20,10, 3]. We used $n/3$ slow modes $\Psi$ to get adjacency matrix $A= \Psi\Psi^T$ used in the GNN layers with output $h^l = \sigma(Ah^{l-1}W +W_s \odot  h^{l-1} + b)$ with self-loop weights $W_s$  and biases $b$ .
> 5. We are improving figure captions.
>
> __2.__ We appreciate your comments and recognize the need for a clear justification of our approach. While our NN model is not a new architecture, using it to map Hessian data to FG modes is a novel approach within physics simulations. This use of NN diverges from traditional ML, mainly centered around supervised and unsupervised learning. We use NN as an ansatz in scientific optimization. This opens up new potential uses of NN in the realm of AI for science. It should also be useful in other ML tasks where saddle points or flat minima are problematic.
>
> Furthermore, our results clearly demonstrate that simpler methods like linear slow mode reparam often fall short compared to GNN. For instance, in Fig. 1 "Pure LJ Loop," where the Lennard-Jones (LJ) potential complicates optimization due to its flatness and shallow minima, traditional methods like GD and linear CG reparam achieve low energies but fail to accurately model the complete coil formation, which the GNN effectively accomplishes. This is evidence for the efficacy of our neural reparam in complex energy landscapes, making our work highly relevant for ML-focused venues.
>
> __3.__ We agree with the need for clearer bio background and will make the following short additions to aid readers:
>
> 1. **Overview of Experiments**: We'll introduce two main experimental setups:
>    - **Energy Minimization on Synthetic Systems**: Forming a coil using LJ potentials to mimic molecular forces.
>    - **MD for Protein Folding**: Using the AMBER force field.
>
> 2. **Details on Synthetic Coil Experiments**:
>    - **Quadratic Bonds + LJ**: We'll describe the energy function $E = E_{bond} + E_{LJ}$ where $E_{bond}$ and $E_{LJ}$ are the quadratic bonds and LJ interactions, designed to form a coil.
>    - **Pure LJ**: Focuses on the challenge of optimizing in a flat energy landscape of LJ, highlighting the difficulty in achieving the lowest energy state due to extremely small gradients.
>
> 3. **Protein Folding Using MD**:
>    - We'll provide a primer on MD's role in studying proteins, emphasizing the use of the AMBER force field to model essential interactions like bond stiffness and angle constraints.
>    - We simplify MD by omitting solvents and focus on energy minimization using AMBER params from OpenMM. This aligns with our synthetic experiments, facilitating interfacing with ML frameworks (pytorch) to implement our approach.
>
> ## Questions
> 1.  As we understand it, metadynamics is mostly concerned with exploring and mapping the free energy landscape, so the goal is different. Nevertheless, our method can be combined with metadynamics by replacing the collective variables with our reparam.
> 2. We recognize the concern. But note that, while slow modes capture low-freq motions and help navigate flat energies, our approach integrates all modes both in the final relaxation phase as well as via residual connections in the GNN, ensuring no critical dynamics are missed.  As in lines 29-33, our motivation for using slow modes is that they converge slowly and that including fast modes forces us to use smaller learning rates for numerical stability. However, the flexibility of our GNN method allows the model to adjustment the weight of fast and slow modes to decrease the energy faster.
>
> 3. Potentially yes, if the slow eigenvectors are transferable. The GNN  graph comes from the Hessian backbone. In some settings such as a protein complex the total Hessian may be approximately block-diagonal, each components forming a block. The slow modes are then mostly localized on individual components. The learned GNN weights (encoding the actual atom locations, for example)  are more challenging to transfer, but the loss should be invariant to symmetries of the system, such as SE(3) for proteins.
> 4. We have not, but that is a good suggestion. Our method can be used with any differentiable potential. We used the empirical AMBER force-field because of its popularity in MD.

---

> > ### Comment · Reviewer_qFP8 · 2024-08-12
> >
> > Thank the author for the response. The rebuttal has addressed most of my concerns. The methods is actually interesting and refreshing for ML research for scientific applications. With that, I will increase my score.

---

> > > ### Author Response · Authors · 2024-08-14
> > >
> > > Thank you very much, we really appreciate your time. You comments really made a difference in our paper's structure and presentation. We think the reparametrization idea is an underexplored aspect of neural nets and that it has a lot of potential, especially as a new class of ansatze for scientific problems. We are also working on generalizability and transferrability of the modes to larger molecules. Fianlly, we are applying this methodology to other systems with metastable and slow dynamics such as glassy systems. We observe in some regimes the method may provide more advantage than in other regimes.

---

### Official Review · Reviewer_ti8n · 2024-07-12

**Soundness:** 3
**Presentation:** 2
**Contribution:** 2
**Rating:** 6
**Confidence:** 3

**Summary:**

This paper proposes an efficient approach for finding the optimal conformation in terms of the energy function. By identifying the slow modes, the proposed method can reduce the computational complexity while extracting the core movement for simulating the dynamics. Specifically, the authors observe a connection between the Laplacian and the Hessian of the targeted potential functions, leading to the coarse-graining by identifying the slow modes.

**Strengths:**

* The proposed method for a proxy of the unweighted adjacency matrix is reasonable and sound.
* The connection between the graph laplacian and the hessian of energy function is interesting and general.
* The proposed method is efficient compared to the gradient descent method which is widely used for simulating the protein dynamics.

**Weaknesses:**

* The efficiency of this work is largely dependent on the complexity of the targeted potential function as it still requires computing the gradient from the potential function. In other words, if we have a large-scale AI model that can more efficiently predict the most stable structures in one shot such as the AlphaFold series, the proposed method might not be helpful to find the optimal conformation.
* The optimization process needs to be clearly described. I suggest the authors provide an algorithm for how to optimize the structures including the intervention of the potential functions and the model architecture.

**Questions:**

* Is it possible to find the globally optimal structures? How much does the initial state affect the final predicted structures?
* For the large-scale protein, how much can the proposed method reduce the computational cost?
* Does the proposed method can significantly reduce the memory cost?

Typos:

line 131: mdoes -> modes

line 176: is relies -> relies

line 176: eighted -> weighted

**Limitations:**

As the authors noted, more experimental results are needed to show the effectiveness of the proposed method. Especially, I recommend comparing the computational cost and the quality of the predicted structures with the state-of-the-art structure prediction models on large-scale proteins.

---

> ### Author Rebuttal · Authors · 2024-08-07
>
> We thank the reviewer for the time taken to review our work. Please check our shared rebuttal above for discussion on improving presentation and new experimental results.
>
> ## Weaknesses:
> 1. __Efficiency:__ We agree that models like AlphaFold take a different approach. However, we think our approach is crucial for domains where we do not have such huge database as proteins. Our approach is inherently "data-free," offering advantages in scenarios where access to molecular dynamics data is limited or unavailable, yet there is a need to simulate the entire system. Drug discovery or material design  would benefit from faster ab initio simulations. This will allow us to go beyond learning patterns in existing molecules and experiment with entirely new ones.
> 2. __Clarity of method:__ Please see our shared response at the top of the page. We have included an enhanced figure in the attached pdf that illustrates the process of extracting coarse-grained modes and their integration into the Graph Neural Network (GNN) reparametrization framework. We also sketch out the GNN architecture in there. This addition aims to clarify the methodology and strengthen the overall presentation of our approach.
>
> ## Questions:
> 1. __global minima and init:__ The energy landscape of proteins is inherently "rugged," (i.e. has many local minima and saddle points) making the search for a global minimum particularly challenging. Consequently, our objective is to identify the most favorable local minima. Even for relatively small proteins, varying initial conditions can result in different local minima. To illustrate how sensitive our methodology is to these initial conditions, we have included a figure in the pdf (see figure c and its caption). For the small protein 2JOF we find starting from three different initially unfolded positions, the final energy obtained by direct gradient descent come out different.
> 2. __Large proteins:__ If the challenges to computing the Hessian for large proteins can be overcome (See our response to rev rxS5, Q3: scaling) our method could save computational costs because it generally requires a fraction of the number of iterations to reach a given energy. (see figure a in pdf). However, each step takes more compute due to forward and backward pass through the NN. With efficient hardware such as GPU, this overhead could reduce significantly.
> 3. __memory cost:__ We think our model actually requires more memory due to the extra neural network. The extra memory cost can be linear or superlinear in the number of particles, depending on how the number of slow modes is chosen to scale with the system size. Our argument is that in the era of large compute, this memory overhead is ok, as the overparametrization can yield significant benefits in terms of convergence to deeper minima and avoiding saddle points.

---

> > ### Comment · Reviewer_ti8n · 2024-08-14
> >
> > Thanks for the authors' clarification. The authors clearly describe their methods by providing additional figures.
> >
> > In my understanding, there are technical pros and cons for this work. For pros, it is data-efficient and requires less memory cost especially compared to the large structure prediction model such as AlphaFold. However, it is disadvantageous in terms of the computation cost of the Hessian which hinders increasing the scalability, and is dependent on the initial state as it can fall into the local minima conformations.
> >
> > On the other hand, in the aspect of the methodology, the connection between the Hessian and Laplacian and identifying effective DOF are promising, which can lead to other future works that are related to protein dynamics.
> >
> > By considering these points, I raise my score to 6

---

> > > ### Author Response · Authors · 2024-08-14
> > >
> > > Thank you very much for your consideration. Yes, we agree with you about the limitations and are working to address that. One observation we had regarding protein dynamics was that, due to the strong quadratic forces from chemical bonds, the slow modes of the Hessian overlapped very highly with the slow modes of the Laplacian of _molecular graph_ alone (we will share the figure soon). This is a big deal, because the molecular graph is very sparse, with average degree between 2-3. Hence, instead of the expensive Hessian computation, we found using the sparse molecule graph led to faster performance. Our final results for protein folding all use the molecule graph as a proxy instead of the Hessian. This makes our method scalable for even large molecules. Additionally, our method could be applied to coarse-grained dynamics and aid CG models find deeper energies.
> > >
> > > We are also actively working on the DOF from Hessian problem and have found further strong results, suggesting one can explore the phase space of molecules efficiently using these DOF. We also agree that it merits its own paper.

---

### Official Review · Reviewer_rxS5 · 2024-07-13

**Soundness:** 2
**Presentation:** 1
**Contribution:** 3
**Rating:** 5
**Confidence:** 3

**Summary:**

This paper presents a novel approach to molecular simulations using neural network reparametrization as an alternative to traditional coarse-graining methods. The key idea is to reparametrize fine-grained modes as functions of coarse-grained modes through a neural network, maintaining continuous access to fine-grained modes and eliminating the need for force-matching. The authors demonstrate improved performance on Lennard-Jones potentials and protein folding simulations compared to conventional methods.

**Strengths:**

* Theoretical foundation: The paper provides a solid theoretical analysis of the properties of physical Hessians and how they relate to slow modes in the system.
* Experimental results: The approach shows promising results on both synthetic systems (Lennard-Jones potentials) and protein folding simulations, demonstrating faster convergence and lower energy states in many cases.

**Weaknesses:**

* Unclear presentation: The paper lacks a clear problem definition and objective function. A flowchart or algorithm illustrating the coarse-graining process, GNN structure, and how it drives molecular dynamics would significantly improve clarity.
* Insufficient explanation of DOF experiments: The paper would benefit from an algorithm or graphical illustration explaining the process of finding effective degrees of freedom.
* Inadequate explanation of GNN graph structure: The authors could have done a better job explaining the graph used in the GNN, particularly the force constant matrix and its physical meaning.
* The paper would benefit from a more extensive comparison to state-of-the-art coarse-graining and optimization methods. Additionally, a more thorough analysis of the computational costs and scalability of the approach would strengthen the paper.

Minor Issues:
* Typos on Line 131, 176

**Questions:**

* What is $n_0$ in line 235?
* Does the design of your GNN need to consider equivariance such as SE(3)? i.e. if you rotate your system, the Hessian and the Laplacian would be transformed in a predicitable way.
* How well do you expect this method to generalize to more complex force fields or larger biomolecular systems?

**Limitations:**

See above.

---

> ### Author Rebuttal · Authors · 2024-08-07
>
> Thank you for your comments. We try to address them below. Please also read our general rebuttal above, which details our plan for improving presentation and more.
>
> ## Weaknesses:
> 1. __Unclear presentation:__ We agree. Please see above and the attached pdf for flow-chart and algorithm.
> 2. __Effective DoF:__ we will add more details, but the gist of it is using Corollary 2.3.1 and 2.3.2 as a loss function to find the symmetry operators $L$. We can approximately satisfy 2.3.1 when $\|LH\|< \epsilon$, which happens when $L$ lives in the slow mode subspace. Next, 2.3.2 can be satisfied when $[L,H]=0$, so we can solve an optimization problem $L_0 = \mathrm{argmin}_L \|[L,H]\|$.
> 3. __GNN graph:__ please see attached pdf and general comment above. The adjacency of the graph (nxn with n being number of particles) is $A = \Psi\Psi^T$ with $\Psi$ being the slow modes.
> 4. __Comparison with SOTA CG:__ True, but unfortunately existing MD frameworks are very opaque and difficult to modify. We tried and failed to interface our method with OpenMM. This seems to be general problem with the field of MD. However, we recently learned about an endeavor to implement MD in Jax. We will try to migrate to that. However, currently, we are unable to offer better comparisons with CG. We decided to present the approach inspite of lacking CG comparison because it is a flexible alternative to CG and can even be combined with CG.
>
> ## Questions:
> 1. __n0__ In the GNN reparam, n0​ denotes the node feature dimension of $Z_h$, which is the input to the GNN, thus n0 is also the input dim of GNN. Recall, the atom positions $X$ are reparametrized as $X = \rho(Z)= GNN_\theta (Z_h)$ (as in equation 2), where $Z=(Z_h, \theta)$ and $\theta$ are the GNN weight, with n0​ being the input hidden dimension.
> 2. __Equivariance:__ very good question! Yes! The formula for the Hessian backbone (eq 7) takes a norm over spatial indices and is therefore invariant under SE(3). We will emphasize this point in the paper. Thus, the Laplacian is also SE(3) invariant and the slow modes do not have a spatial index (scalars under SE(3)).
> 3. __Scaling:__
> Our methodology is compatible with any differentiable potential energy function. The primary limitation for large biomolecular systems is computing the Hessian matrix to get the slow modes. In biomolecules and MD the loss is generally the sum of pair-wise or triple interactions and there fore the Hessian can also be written as the sum of such easy-to-compute Hessians. Thus we don't have to compute the full Hessian matrix until we want the spectrum. We have relied on existing methods such as Lanczos and Davison to get the slow spectrum. Whether the complexity of the force field becomes a burden for scaling depends on the computation graph of the Hessian from autograd, but our guess is that it won't be a problem.
> The GNN can be scaled up by sparsifying the adjacency or other methods people have developed, so we don't expect the reparamterized part to be an issue.

---

> > ### Comment · Reviewer_rxS5 · 2024-08-08
> > **Replies to Rebuttal**
> >
> > I acknowledge that I have read the rebuttal. I ll maintain my score.

---

> > > ### Author Response · Authors · 2024-08-14
> > >
> > > Thank you, we appreciate your time. We wanted to add a couple more points:
> > > * __Scaling for biomolecules:__  We forgot to mention, the method is already very scalable, as follows. We have observed that the Hessian slow modes overlap strongly with the slow modes of the Laplacian of the _molecule graph_. This is likely because the molecule enters the Hessian via quadratic bond energies. We found that just using the full molecule graph inside the GNN, instead of the Hessian slow modes, also yielded good results. Since the molecule graph is quite sparse, and the number of edges scale linearly with the nodes (average degree 2-3), passing through the GNN becomes linear in $n$, making it quite scalable.
> > > * __Equivariance:__ The slow modes can also be made equivariant to other spatial group by using the correct invariant metric. Ex: The Lorentz group $SO(1,3)$ from special relativity preserves the Minkowski metric $\eta = \mathrm{diag}(-1,1,1,1)$, meaning $g^T\eta g = \eta$ for $g\in SO(1,3)$. If instead of the $L_2$ norm on spatial indices we contract them with this metric  $\mathbf{H} \_{ij} = \sum_{\mu\nu\rho\sigma} H_{ij}^{\mu\nu}H_{ij}^{\rho\sigma} \eta_{\mu\rho}\eta_{\nu\sigma}$, we get equivariance under the Lorentz group.
> > >
> > > We hope you also find that the flow-chart and GNN structure plots address your concern and that you still consider the work for a higher score.

---

### Official Review · Reviewer_fqNB · 2024-07-14

**Soundness:** 3
**Presentation:** 2
**Contribution:** 3
**Rating:** 6
**Confidence:** 3

**Summary:**

This paper proposes a novel approach for molecular simulations using neural network reparametrization. The authors first motivate the need for this work, specifically the traditional coarse-graining (CG) methods reduce the number of degrees of freedom (DOF) to improve computational efficiency. However, they require back-mapping and force-matching steps, which can be cumbersome.

 The major contribution is the framework of hessian backbone that allows calculation of hessian using weighted graph Laplacian(although restricted to pairwise invariant potentials). With GNN reparameterization the experiments on synthetic coil shows good improvement in speedup, but not necessarily in reaching lower energies.  The proposed system instead of reducing DOF, allows for a flexible representation of the system. Their  neural reparametrization approach is not limited to reduce DOF but can also increase them when required.

The paper showcases the effectiveness of the method on LJ systems and protein folding simulations. Results suggest the reparametrization approach, especially using GNNs, can achieve ower energy states compared to traditional CG methods and also faster convergence.

**Strengths:**

1) Proposed a hessian backbone approach to get slow modes using graph Laplacian
2) Eliminates the need for force matching and back mapping by reparameterization using slow modes and GNN
3) Average over perturbed configurations taken to address dynamic variation in Hessian
4) Data-free optimization: Doesn't require extensive training data, unlike traditional machine learning approaches.
5) Code is also provided.

**Weaknesses:**

1) Different choices for fraction of eigenvectors in CG equations are mentioned 3x (#AminoAcids), 30%, 50%, and 70%, but the results corresponding to them are not shown, it is important as it is related to the 'epsilon' in slow mode calculations.

2) Literature review is weak, several works both recent and classical on optimization in molecular systems are missing e.g. learned optimizers[1], graph reinforcement learning [2], FIRE [3]

4) Presentation issues:  what is n and d in line 85? The paper can be made more reader friendly if there is a table which has symbol/variable names and their meaning.

Missing citations:
line 56-59
1. Traditional optimization in physics-based models, like (MD), faces unique challenges due to the
shallow nature of these models, where physical DOF are the trainable weights. Additionally, the
interactions occur at multiple scales, from strong covalent bonds to weak van der Waals forces,
leading to slow convergence in gradient-based method

The authors should cite relevant papers for the above paragraph.



[1]Merchant, A., Metz, L., Schoenholz, S.S. &amp; Cubuk, E.D.. (2021). Learn2Hop: Learned Optimization on Rough Landscapes. <i>Proceedings of the 38th International Conference on Machine Learning</i>, in <i>Proceedings of Machine Learning Research</i> 139:7643-7653 Available from https://proceedings.mlr.press/v139/merchant21a.html.

[2]Bihani, V., Manchanda, S., Sastry, S., Ranu, S. &amp; Krishnan, N.M.A.. (2023). StriderNet: A Graph Reinforcement Learning Approach to Optimize Atomic Structures on Rough Energy Landscapes. <i>Proceedings of the 40th International Conference on Machine Learning</i>, in <i>Proceedings of Machine Learning Research</i> 202:2431-2451 Available from https://proceedings.mlr.press/v202/bihani23a.html.

[3]Bitzek, E., Koskinen, P., Gahler, F., Moseler, M., and Gumbsch, P. Structural relaxation made simple. Physical review letters, 97(17):170201, 2006.

**Questions:**

Questions
1) How are the inputs to GCN network 'Z_h0' initialized?
2) Could you clarify what is the loss function used to train the GNN network? details related to training shall be provided.
3) Can the approach be used for disordered glassy systems, e.g. Kob-Anderson Binary LJ Model Glass which is known to have slow dynamics. Does it gets stuck in higher energy minima? It will be good to show the coarse grained approach on glassy systems.
4) The authors mention they use Adam optimizer with a learning rate 10−2. Can the authors show impact of learning rate, lower and high Learning rates? Does the conclusion remain same?

Minor Comments

1) Typo in line 143: Power of z in repulsive term
2) Typo in line 176: spelling of 'weighted'
3) Line 131 typo: mdoes

**Limitations:**

Yes

---

> ### Author Rebuttal · Authors · 2024-08-07
>
> Thank you, we appreciate your pertinent comments. Please also check our general response above about improvements to the presentation.
> ## Weaknesses:
> 1. The plot was mistakenly omitted. We are adding it to the appendix.
> 2. Thanks, adding the citations.
> 3. n is the number of particles, d is the embedded dim, e.g. d=3 for 3D coordinates x,y,z.
> 4. Missing citations: we are adding the citations for that paragraph
>
> ## Questions:
> 1. For $Z_{h0}$ initialization, we used two methods: 1) random; 2) started from random but optimized together with GNN weights to match a given initial configuration of atoms. Method 2 introduces a minimal overhead: for protein 2JOF with 284 atoms it converges after 900 steps, taking 0.67 seconds.
> 2. The GNN loss is the energy function $\mathscr{L} = E$! This is the crucial point about reparametrization. We still deal with $E(X)$ but now $X$ is $X = \rho(Z)$ (as in eq 2) where $Z=(Z_h, \theta)$ and $\theta$ are the GNN weights. So the optimization problem changes from $\mathrm{argmin}_X E(X)$ to $\mathrm{argmin}_Z E(\rho(Z))$ for the GNN or other reparametrizations.
> Our framework is unsupervised. The training of the GNN is the goal: The positions of atoms are encoded in GNN weights and finding the final atom positions means minimizing the loss function which is the potential energy function. After the training, there is no inference step.
> Our GNN setup consists of two GCN layers with residual connections, followed by a projection to 3D. $Z_h$ and each GCN layer comprises 100 hidden dimensions. During training, once the energy curve begins to plateau (a minimum change in energy of 0.1 and 20 patience steps), we stop the GNN reparam. We then continue to optimize using full FG modes again using the same loss (energy) function as GNN.
> 3. This is a great suggestion. We believe that the Kob-Andersen Binary Lennard-Jones (LJ) glass model (KAM) could also be a potential application. KAM closely resembles our synthetic pure LJ loop simulations, with the key difference being that, in KAM, particles do not have a fixed underlying graph during optimization. Consequently, in KAM, we need to construct a spatial proximity graph and update it dynamically during training to leverage the GNN framework. Due to time constraints, we haven't conducted extensive experiments, but we have run a few, both a gradient descent (GD) and a GNN reparam.
> 4. Yes, we have run sweeps of the learning rate and we find the lowest energy is consistently achieved by a GNN, though direct GD beats some GNN that have large LR.

---

> > ### Comment · Reviewer_fqNB · 2024-08-10
> > **Thanks**
> >
> > I thank the authors for their response. After these clarifications, I increase my score.
> >
> > I agree with other reviewers, the presentation needs to be improved. Also, the new figure with learning rate sweep could be a continuous plot(instead of the discrete version), showing change in energy.

---

> > > ### Author Response · Authors · 2024-08-14
> > >
> > > Thank you, we will also produce the loss curve plots in the coming days. We will also try to include our preliminary results on the Kob-Anderson system. We are definitely planning to apply this method to glassy systems. It would be interesting to investigate if in certain hard regimes this methodology would provide more advantage than in other regimes.

---

### Author Rebuttal · Authors · 2024-08-06

Thank you all for very constructive comments. Some issues were raised by multiple reviewer. Here we will address the shared concerns. Below, we respond point-by-point to each review.

## Presentation
We agree that the presentation of the paper needs significant improvement and reorganization, especially intro and background. We are doing the following:
1. __Problem statement:__ Our revised intro will start with an overview of challenges in scientific simulations, such as the proliferation of saddle points and local minima leading to suboptimal results. While conventional dim reduction methods like CG offer partial solutions, they encounter issues like back-mapping and force-matching. Instead, we propose an innovative approach using an overparametrized neural ansatz. We demonstrate that CG reparam or a well-designed GNN ansatz, incorporating Hessian slow modes, achieve significantly lower energy states compared to direct optimization.
2. __Flow-chart and algorithm:__ We are making a new Fig. 1 (see attached pdf) to outline the methodology. We are also adding the step-by=step of the algorithm (detailed next).
3. __Motivation for using Hessian slow-modes:__ In Sec. 2 on the Hessian, we'll clarify the motivation for using slow modes, rooted in the difference in fast vs slow mode evolution rate, which causes slow convergence at saddle points. Our goal is to adapt the optimization process to grant direct access to slow modes, hoping that it helps escape such saddle points. However, this approach faces challenges: 1) Changes in the Hessian may alter the slow modes during optimization. 2) The need to modify the optimization to favor slow modes. We address these by showing the stability and robustness of slow modes and by proposing linear CG and GNN reparam. Our experiments show superior efficacy of the GNN approach.
4. __GNN details:__ We add GNN parameter details in a table in the appendix. Our GNN consists of GCN layers with self-loops and residual connections. For the experiments in Fig 1, the GNN hidden dims are [20,10, 3], i.e. starting from $Z_h$ with 20 dim embedding, and one GCN layer with 10 hidden dims and a projection layer down to 3D. For the protein experiments we had dims [100,100,3]. We used $n/3$ slow modes $\Psi$ to get adjacency matrix $A= \Psi\Psi^T$ used in the GNN layers, which are GCN with output $h^l = \sigma(Ah^{l-1}W +W_s \odot  h^{l-1} + b)$ with self-loop weights $W_s$  and biases $b$ .   Here $h^l \in \mathbb{R}^{n\times d_l}$ ($n$ particles in $d_l$ hidden dims). We concatenate outputs along feature dims of all GCN layers into $h = [Z_h| h^1...]$ (dense residual connections) and pass them through a final projection along feature dims to get them to $d=3$ dimensions.
5. __Figures:__ We are improving figure captions. We are also including missing figures on using different number of Hessian modes (30%-70%) for proteins.
6. __More experiments:__ We are adding new figures comparing our final energies with OpenMM’s `simulation.minimizeEnergy` which is the most head-to-head comparison with our method. However, we OpenMM also uses many tricks for efficiency, such as distance cutoff for forces and may use Barnes-Hut or other space partitioning, which we haven't implemented. Thus the only fair comparison we can make right now is against direct gradient descent (no reparam). We run a __learning rate sweep__ for some protein simulations. We find the lowest loss and closest to OpenMM minimum to be consistently GNN (see attached pdf). We also find that at a similar number of iteration steps (5k) OpenMM was at much higher energies compared to the GNN.
## Flow-chart and algorithm
Most of the reviewers asked for a clear algorithm or flow-chart of our methodology. We are including the flow-chart in the attached pdf and will add the algorithm steps as follows:
1. Compute functional Hessian of the energy function $H = \nabla \nabla \mathscr{L}$ w.r.t. Particle positions $X$. Because $X \in \mathbb{R}^{n\times d}$ ($n$ particles with $d$ features), this Hessian will have four indices as $H_{i j}^{\mu\nu} = \partial^2 \mathscr{L}/\partial X_i^\mu \partial X_j^\nu$
2. Evaluate the Hessian over a small ensemble of perturbed positions $\mathbf{Samples}(X) = \{X’ = X+\delta X\}$ and compute the __Backbone__  $\mathbf{H} \_{ij}$ $= \sum_{X’\in \mathbf{Samples}(X)}\sum_{\mu\nu} H_{ij}^{\mu\nu}(X’)^2$(eq 7).
3. Compute or approximate $k$ eigenvectors $\Psi$ of $\mathbf{H}$ with smallest magnitude eigenvalues.
4. Perform neural reparametrization: express $X$ using a neural ansatz $X = \mathrm{NN}(\Psi,\theta)$ where $\mathrm{NN}$ uses the Hessian modes $\Psi$ and has trainable parameters $\theta$.
5. Optimize the same los $\mathscr{L}(X)= \mathscr{L}(\mathrm{NN}(\Psi,\theta))$ over NN parameters $\theta$ instead of the original $X$.

The procedure is quite simple and could be summarized in the following pythonic pseudocode using pytorch (actual implementation slightly different for efficiency):

```python
# Hessian backbone
H = functional.hessian(Loss)
H_samples = tensor([H(x) for x in samples])
H_backbone = H_sampels.norm((2,4)).sum(0)
H_lap = Laplacian(H_backbone)

# slow modes
eig_vals, Psi = eigh(H_lap)
Psi_slow = Psi[:, argsort(abs(eig_val))[:k]]

# reparametrization
A = Psi_slow @ Psi_slow.T
gnn = GNN(A, hidden_dims = [h,h,3])
Z = random.normal(n,h).requires_grad()
X = gnn(Z)

# optimization
optimizer = Adam(parameters = [Z]+list(gnn.parameters())

for i in range(steps):
	optimizer.zero_grads()
	loss = Loss(X)
	loss.backward()
	optimizer.step()

```

---

### Comment · Area_Chair_u9qD · 2024-08-13
**Request to reviewers to respond to rebuttal comments**

Dear Reviewers,

We are into the last 24 hours of author-reviewer discussions. As you might have noticed, authors have actively tried to engage in the discussions. Hence, I request all the reviewers to please respond to the rebuttal at the earliest so that the authors get a fair chance to represent themselves.

Thank you,

AC

---

### Decision · Program_Chairs · 2024-09-25

**Decision:**

Accept (poster)

**Comment:**

Accelerating molecular simulations is an important problem in several scientific domains. In this work, authors propose a novel approach to address this challenge through neural reparametrization. All the reviewers have appreciated the proposed framework and its relevance. However, they have also raised concerns regarding the clarity of presentation, setting up the problem, improved comparison with the state of the art and more extensive literature review. Authors have satisfactorily addressed these comments. It is recommended that all these changes, along with the modifications to reflect the review responses, are implemented in the camera ready version of the manuscript.